# Hold Onto That Thought: Assessing KV Cache Compression On Reasoning

## Abstract

Large language models (LLMs) have demonstrated remarkable performance on long-context tasks, but are often bottlenecked by memory constraints. Namely, the KV cache, which is used to significantly speed up attention computations, grows linearly with context length. A suite of compression algorithms has been introduced to alleviate cache growth by evicting unimportant tokens. However, several popular strategies are targeted towards the prefill phase, i.e., processing long prompt context, and their performance is rarely assessed on reasoning tasks requiring long decoding. In particular, short but complex prompts, such as those in benchmarks like GSM8K and MATH500, often benefit from multi-step reasoning and self-reflection, resulting in thinking sequences thousands of tokens long. In this work, we benchmark the performance of several popular compression strategies on long-reasoning tasks. For the non-reasoning Llama-3.1-8B-Instruct, we determine that no singular strategy fits all, and that performance is heavily influenced by dataset type. However, we discover that H2O and our decoding-enabled variant of SnapKV are dominant strategies for reasoning models, indicating the utility of heavy-hitter tracking for reasoning traces. We also find that eviction strategies at low budgets can produce longer reasoning traces, revealing a tradeoff between cache size and inference costs.

## 1 Introduction

Large language models (LLMs) have demonstrated remarkable performance on complex NLP tasks that require multi-step reasoning. Unlike summarization tasks (Bai et al., 2023; Fabbri et al., 2021) and keyword tracking tasks (Hsieh et al., 2024), which scale task complexity with context length, reasoning benchmarks challenge models to generate answers that are not clearly contained in the prompt. Such tasks include reading comprehension (Dua et al., 2019; Yu et al., 2020), commonsense reasoning (Zellers et al., 2019; Talmor et al., 2018; Geva et al., 2021), first-order logic (Han et al., 2022; Kwon et al., 2025), and mathematical problem-solving (Cobbe et al., 2021).

Reasoning benchmarks differ from long-context tasks in that they normally compel the LLM to provide answers that are longer than the question itself. This can pose a serious resource problem for the LLM, as past token key and value embeddings are maintained in memory to avoid redundant attention calculations. This key-value (KV) cache grows linearly with sequence length, which can result in memory blowup for older or single-GPU setups. Furthermore, specialized reasoning models such as DeepSeek-R1 (Guo et al., 2025) and the Llama-Nemotron series are known to output excessively long reasoning traces (Cai et al., 2025; Fatemi et al., 2025) which outnumber the length of the prompt itself by hundreds to thousands of tokens.

To address the memory demands of long sequences, numerous KV cache compression methods have been proposed. These techniques generally maintain a fixed KV cache size by selectively discarding tokens deemed "unimportant". However, defining token importance" is non-trivial, and different approaches rely on distinct heuristics: attention scores (Zhang et al., 2023; Liu et al., 2023; Li et al., 2024), cosine similarity (Liu et al., 2024a; Han et al., 2023), embedding norms (Devoto et al., 2024), and head-specific token-type preferences (Ge et al., 2023). Despite this variety, most evaluations of cache compression have focused on long-context benchmarks such as LongBench (Bai et al., 2023) and RULER (Hsieh et al., 2024), or on heterogeneous batteries like LM Eval Harness ((Gao et al., 2024)), rather than tasks where the generation length, not the prompt, dominates memory usage.

In this work, we conduct a comprehensive assessment of the major state-of-the-art KV cache compression strategies across eight reasoning benchmarks: FOLIO (Han et al., 2022), DROP (Dua et al., 2019), GSM8K (Cobbe et al., 2021), MATH-500 (Lightman et al., 2023), ReClor (Yu et al., 2020), StrategyQA (Geva et al., 2021), CommonSenseQA (Talmor et al., 2018), and OpenBookQA (Mihaylov et al., 2018). Together, these benchmarks span four critical reasoning categories: reading comprehension, common sense, logical reasoning, and mathematical reasoning. We evaluate these strategies on Llama-3.1-8B-Instruct as well as four reasoning models: Llama-3.1-Nemotron-Nano-8B-v1, DeepSeek-R1-Distill-Llama-8B, and DeepSeek-R1-Distill-Qwen-7B/14B. By focusing on long-generation rather than long-prompt scenarios, our study fills a notable gap in the existing literature. Our primary contributions are threefold:

***A comprehensive benchmark:*** We conduct a comprehensive evaluation of major KV cache compression strategies, including StreamingLLM, H2O (Zhang et al., 2023), a decoding-enabled SnapKV (Li et al., 2024), R-KV (Cai et al., 2025), and KNorm (Devoto et al., 2024), across a suite of eight benchmarks spanning mathematical, logical, and commonsense reasoning. We evaluate over several realistic settings, cache, and max token budgets for a single-GPU system.

***Renewed attention for attention-based compression:*** Our analysis reveals that classical attention-based "heavy-hitter" strategies, which evict tokens based on accumulated attention scores, significantly outperform other methods, even defeating full-cache reasoning occasionally. Namely, this includes H2O and our novel and simple extension of SnapKV (prompt-only compression method) to a decoding-enabled variant, SnapKV-**D**ecoding. Both methods, especially SnapKV-D, win over *all* budgets and datasets for reasoning models.

***A library for analyzing decoding compression:*** We implement a fork of the NVIDIA `kvpress`[1] library, which adds support for decoding phase compression for any kvpress method. We add support for R-KV and H2O to the kvpress. Our goal is to provide an open-source playground for analyzing end-to-end KV cache compression strategies.

## 2 PRELIMINARIES

In this section, we briefly review the concepts of large language models, LLM inference and autoregressive generation, the KV cache, and the chain-of-thought (CoT) reasoning.

**Transformer Architectures and Autoregressive Generation.** Modern Large Language Models (LLMs) predominantly operate as autoregressive, decoder-only Transformers (Vaswani et al., 2017; Radford et al., 2019; Achiam et al., 2023; Touvron et al., 2023). This architecture generates text sequentially, producing one token at a time by conditioning on the entire preceding sequence of tokens, which includes both the initial prompt and any previously generated output (Brown et al., 2020). Importantly, the model's ability to maintain coherent and contextually relevant generation over time is crucial to its capabilities, especially in tasks requiring reasoning or narrative development (Lee et al., 2024; Zhang et al., 2025).

**Self-Attention Mechanism and the KV Cache Bottleneck.** During generation, a query ($q$) vector for the current token is *attends* to a series of Key ($k$) and Value ($v$) vectors corresponding to every token in the preceding context. In this process, notably, for the generation of every new token, the entire sequence of Key and Value vectors for *all* previous tokens should be accessed. To avoid recomputing these K-V pairs at each step, they are stored in the Key-Value (KV) cache, the size of which grows linearly with the sequence length ($n$), resulting in an $O(n)$ memory complexity that creates a significant bottleneck. Formally, for a sequence of $n$ tokens, we denote the query cache $Q_l^h \in \mathbb{R}^{n \times d}$, key cache $K_l^h \in \mathbb{R}^{n \times d}$, and value cache $V_l^h \in \mathbb{R}^{n \times d}$, where $d$ is the embedding dimension, $l$ is the layer, and $h$ denotes a head for multi-head attention layers (Vaswani et al., 2017). The dot-product self-attention mechanism is defined as $A_l^h(Q_l^h, K_l^h, V_l^h) = \text{softmax}(Q_l^h(K_l^h)^\top/\sqrt{d})V_l^h$. To avoid linear scaling with sequence length, *token eviction* methods, the key focus of work, discard embeddings of previous tokens which are no longer "important" to the current decoding step.

---

[1]https://github.com/NVIDIA/kvpress

**Objective.** A common objective to study the quality of an importance heuristic is to minimize the deviation between the outputs of a non-evicted and evicted attention layer. More specifically, let $\bar{K}_l^h$ and $\bar{V}_l^h$, respectively, denote an evicted key and value cache. We may interpret these caches as sparse matrices by dropping all but $B$ rows (the budget) of $K_l^h$ and $V_l^h$. Attention is typically followed by multiplication with an output projector $W_O \in \mathbb{R}^{d \times p}$ and passage through a 2-layer MLP $\mathcal{F}(x) := x + W_2\mathrm{relu}(W_1x)$, where $W_1, W_2$ are trained hidden weights. Let $x = A_l^h(Q_l^h, K_l^h, V_l^h)$ and $\bar{x} = A_l^h(Q_l^h, \bar{K}_l^h, \bar{V}_l^h)$. The objective of any KV eviction algorithm is to minimize $\mathbb{E}[||\mathcal{F}(x) - \mathcal{F}(\bar{x})||_2]$, where randomness is with regards to the attention distribution. Guarantees on this error are scarce in the eviction landscape, with the most prominent presented for heavy-hitter Scissorhands approach in Liu et al. (2023), which asserts an upper bound that scales with $1 - B/N$, where $N$ is the sequence length and assumes a heavy-tailed distribution of attention scores, which is frequently observed (Devoto et al., 2024; Liu et al., 2024a; 2023). However, many popular eviction strategies, such as StreamingLLM, KNorm, and PyramidKV are based on empirical observations as opposed to attention-tracking and thus are currently not guaranteed.

**Chain-of-Thought and Multi-Step Reasoning.** While many long-context applications involve processing long prompts, a critical class of tasks requires long-form generation from short and complex prompts. Prompting strategies such as Chain-of-Thought (CoT) encourage models to externalize their reasoning process, generating intermediate "thinking" steps that can extend for hundreds or thousands of tokens to solve a problem (Wei et al., 2022; Wang et al., 2022). Benchmarks such as GSM8K (Cobbe et al., 2021) are representative of this domain, where the path to the correct answer necessitates a lengthy, self-generated chain of reasoning.

## 3 RELATED WORK

### 3.1 KV CACHE COMPRESSION

KV cache compression is a rich field of study composed of strategies ranging from quantization (Hooper et al., 2024; Ashkboos et al., 2024; Liu et al., 2024b) to offloading methods that move the entire cache to the CPU which is significantly less memory bound (Sun et al., 2024; Chen et al., 2024; Tang et al., 2024). However, in this work, we are focused on strategies which maintain a constant cache size, thus permitting arbitrary generation length.

#### 3.1.1 TOKEN EVICTION

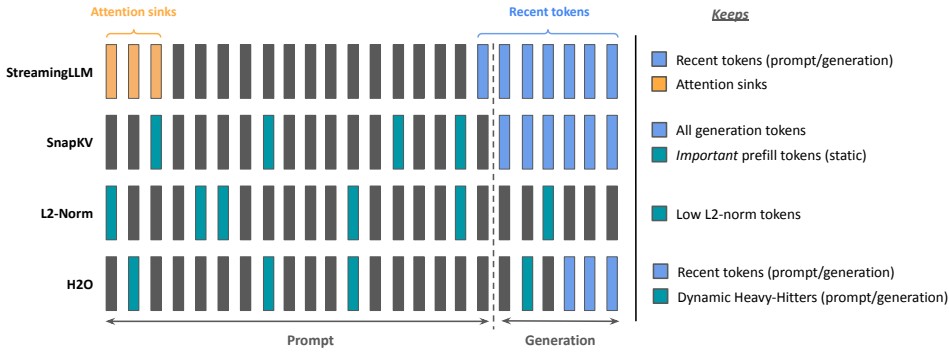

Figure 1: A Conceptual Comparison of Token Retention Strategies in Different KV Cache Compression Methods. Each row illustrates a method's logic for retaining tokens (colored) versus evicting them (gray) from the KV cache during a long sequence divided into a prefill and decoding phase.

A primary line of research for mitigating the memory burden of the KV cache involves *token eviction*. These methods aim to reduce the cache size by selectively removing or merging tokens deemed less important. To achieve this, multiple approaches have been developed, including recency-based approaches such as simple sliding window (Beltagy et al., 2020), importance-based methods that retain "attention sinks" or heavy-hitter tokens from the prompt (Xiao et al., 2023; Zhang et al.,

2023; Li et al., 2024; Liu et al., 2023), dynamically adjustment of KV caches per layer for optimal efficiency-utility balancing (Cai et al., 2024), redundancy-aware techniques that merge semantically similar states (Cai et al., 2025). Figure 1 provides a conceptual comparison of the most important approaches we cover in this work.

**StreamingLLM's** (Xiao et al., 2023) strategy is to always maintain the KV states of the first few (e.g., four) tokens, which serve as the attention sinks, and combine them with a sliding window of the most recent tokens (up to the available budget). **H2O** (Zhang et al., 2023) dynamically identifies important or "heavy hitter" tokens based on their cumulative attention scores received during generation. The H2O cache is composed of two parts: a budget for the most recent tokens and a budget for the H2 tokens. **SnapKV** (Li et al., 2024) focuses primarily on compressing the KV cache of the initial prompt during the prefill stage. SnapKV uses a small "observation window" at the end of the prompt to predict importance. The attention scores from queries in this observation window are aggregated to "vote" for important positions (heavy hitters) in the prefix. R-KV (Cai et al., 2025), designed for reasoning trace compression, uses a combination of accumulated attention score and pair-wise key cosine similarities to identify unimportant tokens. A distinct and computationally efficient approach, which we refer to as the **KNorm** strategy (Devoto et al., 2024), bypasses the need for attention scores entirely. Specifically, the authors observe that tokens whose key vectors have a low $L_2$ norm consistently attract high attention scores from subsequent queries.

### 3.2 BENCHMARKING REASONING

**GSM8K** (Grade School Math 8K) is a widely-used dataset of grade-school level math word problems (Cobbe et al., 2021). More advanced challenges are drawn the **MATH-500** dataset (Lightman et al., 2023), which contains competition-level problems across algebra, geometry, and number theory. **ReClor** (Yu et al., 2020) is a reading comprehension dataset built from GMAT and LSAT logical reasoning questions. Similarly, **LogiQA** (Liu et al., 2020) provides multiple-choice questions from civil service exams that require a deep understanding of logical puzzles and deductions. For evaluating capabilities in more formal systems, the **FOLIO** (Han et al., 2022) dataset assesses natural language reasoning in the context of First-Order Logic (FOL). Beyond formal and mathematical logic, a significant portion of research focuses on commonsense reasoning. **StrategyQA** (Geva et al., 2021) tests a model's ability to infer the implicit reasoning steps needed to answer a yes/no question by asking for the underlying strategy. Another tested benchmarks is **CommonsenseQA**, which tests a model's ability to reason with general world knowledge. Finally, the integration of textual understanding with quantitative skills is measured by benchmarks such as **DROP** (Dua et al., 2019). This reading comprehension dataset is unique in that answering its questions requires performing discrete operations like counting, sorting, or simple arithmetic directly on the information presented.

## 4 EXPERIMENTS & ANALYSIS

### 4.1 SETUP

*KV Compression Methods.* We test **H2O**, **R-KV**, **StreamingLLM**, **KNorm**, our own decoding-variant of SnapKV which we call **SnapKV-D**, and **ShadowKV** (Sun et al., 2024). We note that ShadowKV uses the CPU to offload the cache and thus is not a true compression strategy. However, offloading strategies represent an important class of compression methods; thus, we include them as a baseline. For SnapKV-D, we now allow the observation window to slide along the decoded sequence at regular intervals (every window size $w = 128$) steps. This is detailed further in Appendix 10. Further hyperparameter details are described in Appendix A.11.

*Models.* We test the base, non-reasoning Llama-3.1-8B-Instruct and three reasoning models: DeepSeek-R1-Distill-Qwen-7B/14B, Nemotron-Nano-8B-v1 and DeepSeek-R1-Distill-Llama-8B.

*Datasets.* We divide our benchmark into 4 distinct groups: **(1) Reading Comprehension:** DROP, ReClor; **(2) Logical Reasoning:** StrategyQA, FOLIO; **(3) Commonsense Reasoning:** OpenBookQA (OBQA), CommonsenseQA (CSQA); **(4) Math Reasoning:** MATH-500, GSM8K. For each dataset, we randomly sample 100 questions for two different seeds.

*Performance.* For benchmarking the individual compression strategies, we use the NVIDIA kvpress library, which natively provides most of the targeted algorithms. We provide each dataset to each

Table 1: **Llama-3.1-8B-Instruct**. Varying compressions and budgets on a *non-reasoning* model.

| Llama-3.1-8B-Instruct | GSM8K | | | | Math500 | | | | CSQA | | | | OBQA | | | |
|---|---|---|---|---|---|---|---|---|---|---|---|---|---|---|---|---|
| | 128 | 256 | 384 | 512 | 128 | 256 | 384 | 512 | 128 | 256 | 384 | 512 | 128 | 256 | 384 | 512 |
| Full | 0.88 | | | | 0.39 | | | | 0.77 | | | | 0.84 | | | |
| ShadowKV | 0.32 | | | | 0.22 | | | | 0.20 | | | | 0.31 | | | |
| H2O | **0.63** | **0.77** | 0.82 | 0.83 | **0.30** | **0.33** | **0.33** | **0.36** | 0.74 | 0.76 | **0.77** | **0.77** | 0.83 | 0.86 | 0.86 | 0.86 |
| Knorm | 0.05 | 0.53 | 0.73 | 0.82 | 0.03 | 0.18 | 0.22 | 0.33 | 0.34 | **0.77** | 0.75 | 0.76 | 0.41 | 0.79 | 0.84 | 0.82 |
| RKV | 0.12 | 0.34 | 0.50 | 0.49 | 0.03 | 0.10 | 0.16 | 0.20 | 0.36 | 0.62 | 0.76 | 0.77 | 0.22 | 0.66 | 0.77 | 0.84 |
| SnapKV | 0.53 | 0.55 | 0.56 | 0.53 | 0.20 | 0.21 | 0.19 | 0.20 | 0.70 | 0.64 | 0.71 | 0.72 | 0.73 | 0.77 | 0.72 | 0.76 |
| StreamingLLM | 0.26 | 0.75 | **0.84** | **0.87** | 0.11 | 0.26 | 0.32 | 0.35 | 0.20 | 0.75 | 0.76 | **0.77** | 0.14 | 0.72 | 0.84 | 0.84 |
| | ReClor | | | | DROP | | | | StrategyQA | | | | FOLIO | | | |
| | 128 | 256 | 384 | 512 | 128 | 256 | 384 | 512 | 128 | 256 | 384 | 512 | 128 | 256 | 384 | 512 |
| Full | 0.60 | | | | 0.15 | | | | 0.83 | | | | 0.51 | | | |
| ShadowKV | 0.27 | | | | 0.28 | | | | 0.68 | | | | 0.33 | | | |
| H2O | 0.32 | 0.56 | **0.60** | 0.58 | 0.12 | **0.14** | **0.17** | **0.17** | **0.81** | **0.87** | 0.88 | **0.89** | 0.22 | **0.43** | 0.41 | 0.43 |
| Knorm | 0.01 | 0.19 | 0.46 | **0.59** | 0.01 | 0.08 | 0.13 | 0.13 | 0.47 | 0.85 | 0.88 | 0.87 | 0.02 | 0.28 | 0.39 | 0.38 |
| RKV | 0.04 | 0.21 | 0.40 | 0.54 | 0.06 | 0.07 | 0.14 | 0.11 | 0.60 | 0.79 | 0.77 | 0.79 | 0.07 | 0.36 | 0.44 | 0.34 |
| SnapKV | **0.53** | **0.57** | 0.58 | 0.55 | **0.15** | 0.12 | 0.11 | 0.12 | 0.78 | 0.78 | 0.81 | 0.76 | **0.44** | 0.40 | **0.45** | **0.46** |
| StreamingLLM | 0.05 | 0.21 | 0.59 | 0.58 | 0.09 | 0.11 | 0.15 | 0.16 | 0.11 | 0.76 | **0.89** | 0.85 | 0.03 | 0.09 | 0.25 | 0.35 |

Table 2: **Deepseek-R1-Distill-Qwen-7B.** Performance using varying compressions and budgets.

| Deepseek-R1-Distill-Qwen-7B | GSM8K | | | | Math500 | | | | CSQA | | | | OBQA | | | |
|---|---|---|---|---|---|---|---|---|---|---|---|---|---|---|---|---|
| | 128 | 256 | 384 | 512 | 128 | 256 | 384 | 512 | 128 | 256 | 384 | 512 | 128 | 256 | 384 | 512 |
| Full | 0.70 | | | | 0.47 | | | | 0.67 | | | | 0.78 | | | |
| ShadowKV | 0.47 | | | | 0.33 | | | | 0.20 | | | | 0.31 | | | |
| H2O | 0.21 | 0.44 | 0.51 | 0.52 | 0.14 | 0.21 | 0.29 | 0.31 | 0.44 | 0.61 | **0.60** | **0.64** | 0.42 | 0.64 | **0.69** | 0.67 |
| Knorm | 0.00 | 0.00 | 0.08 | 0.16 | 0.00 | 0.01 | 0.03 | 0.05 | 0.05 | 0.13 | 0.30 | 0.42 | 0.03 | 0.05 | 0.23 | 0.38 |
| RKV | 0.04 | 0.07 | 0.18 | 0.30 | 0.04 | 0.04 | 0.05 | 0.17 | 0.10 | 0.09 | 0.27 | 0.34 | 0.10 | 0.10 | 0.21 | 0.26 |
| SnapKV | **0.67** | **0.67** | **0.70** | **0.71** | **0.38** | **0.36** | **0.36** | **0.32** | **0.65** | **0.62** | 0.59 | 0.61 | **0.71** | **0.75** | 0.68 | **0.76** |
| StreamingLLM | 0.02 | 0.19 | 0.32 | 0.44 | 0.03 | 0.12 | 0.19 | 0.26 | 0.08 | 0.14 | 0.31 | 0.48 | 0.02 | 0.11 | 0.28 | 0.37 |
| | ReClor | | | | DROP | | | | StrategyQA | | | | FOLIO | | | |
| | 128 | 256 | 384 | 512 | 128 | 256 | 384 | 512 | 128 | 256 | 384 | 512 | 128 | 256 | 384 | 512 |
| Full | 0.45 | | | | 0.16 | | | | 0.67 | | | | 0.36 | | | |
| ShadowKV | 0.27 | | | | 0.14 | | | | 0.60 | | | | 0.33 | | | |
| H2O | 0.01 | 0.04 | 0.18 | 0.28 | 0.04 | 0.07 | 0.10 | 0.10 | 0.33 | **0.64** | **0.74** | **0.72** | 0.03 | 0.21 | 0.23 | 0.23 |
| Knorm | 0.00 | 0.00 | 0.01 | 0.01 | 0.00 | 0.01 | 0.01 | 0.03 | 0.00 | 0.12 | 0.44 | 0.59 | 0.00 | 0.01 | 0.03 | 0.05 |
| RKV | 0.04 | 0.03 | 0.02 | 0.01 | 0.04 | 0.04 | 0.03 | 0.04 | 0.05 | 0.14 | 0.34 | 0.42 | 0.04 | 0.03 | 0.02 | 0.06 |
| SnapKV | **0.45** | **0.39** | **0.40** | **0.43** | **0.13** | **0.11** | **0.12** | **0.16** | **0.60** | 0.59 | 0.57 | 0.63 | **0.30** | **0.25** | **0.31** | **0.29** |
| StreamingLLM | 0.00 | 0.01 | 0.01 | 0.04 | 0.04 | 0.05 | 0.08 | 0.13 | 0.00 | 0.05 | 0.22 | 0.42 | 0.00 | 0.01 | 0.02 | 0.03 |

model over the cache budgets {128, 256, 384, 512}. Each model is allowed to generate a maximum of 2048 new tokens via greedy decoding. This token limit is enforced to better simulate a resource-constrained setting for inference and also based on mean generation lengths reported in Table 7.. We use author-recommended hyperparameters for all methods. Accuracy benchmarks were performed on an HPC cluster using an NVIDIA RTX A6000 48GB GPU.

## 4.2 LATENCY EXPERIMENT

Although this benchmark is primarily concerned with accuracy, we assess the latency of our tested methods in Figure 2 and Table 5 to gather a more complete picture of efficiency. StreamingLLM and KNorm do not compute accumulated attention scores thus they incur less overhead than H2O and SnapKV-D. These results concurs with the computational overhead summarized in Table 13.

Table 3: **Nemotron-Nano-8B.** Performance using varying compressions and budgets.

| Nemotron-Nano-8B | GSM8K | | | | Math500 | | | | CSQA | | | | OBQA | | | |
|---|---|---|---|---|---|---|---|---|---|---|---|---|---|---|---|---|
| | 128 | 256 | 384 | 512 | 128 | 256 | 384 | 512 | 128 | 256 | 384 | 512 | 128 | 256 | 384 | 512 |
| Full | 0.64 | | | | 0.45 | | | | 0.51 | | | | 0.64 | | | |
| ShadowKV | 0.44 | | | | 0.28 | | | | 0.20 | | | | 0.31 | | | |
| H2O | 0.22 | 0.45 | 0.52 | 0.57 | 0.16 | 0.24 | 0.31 | 0.33 | 0.47 | 0.49 | **0.52** | 0.51 | 0.59 | 0.59 | 0.58 | 0.62 |
| Knorm | 0.01 | 0.02 | 0.09 | 0.18 | 0.01 | 0.01 | 0.03 | 0.06 | 0.36 | 0.40 | 0.44 | 0.46 | 0.32 | 0.44 | 0.48 | 0.57 |
| RKV | 0.04 | 0.03 | 0.09 | 0.15 | 0.02 | 0.04 | 0.03 | 0.06 | 0.28 | 0.30 | 0.42 | 0.41 | 0.35 | 0.44 | 0.51 | 0.51 |
| SnapKV | **0.65** | **0.63** | **0.66** | **0.66** | **0.41** | **0.44** | **0.45** | 0.43 | 0.49 | **0.50** | 0.51 | **0.53** | **0.68** | 0.63 | **0.66** | **0.66** |
| StreamingLLM | 0.03 | 0.20 | 0.40 | 0.53 | 0.02 | 0.13 | 0.22 | 0.34 | 0.36 | 0.44 | 0.46 | 0.50 | 0.36 | 0.46 | 0.52 | 0.62 |
| | ReClor | | | | DROP | | | | StrategyQA | | | | FOLIO | | | |
| | 128 | 256 | 384 | 512 | 128 | 256 | 384 | 512 | 128 | 256 | 384 | 512 | 128 | 256 | 384 | 512 |
| Full | 0.48 | | | | 0.11 | | | | 0.89 | | | | 0.36 | | | |
| ShadowKV | 0.27 | | | | 0.11 | | | | 0.65 | | | | 0.33 | | | |
| H2O | 0.20 | 0.22 | 0.35 | **0.40** | 0.05 | 0.06 | 0.10 | 0.09 | 0.76 | 0.84 | **0.85** | 0.83 | 0.22 | 0.36 | 0.35 | 0.37 |
| Knorm | 0.01 | 0.03 | 0.07 | 0.07 | 0.01 | 0.01 | 0.02 | 0.03 | 0.38 | 0.55 | 0.68 | 0.76 | 0.03 | 0.04 | 0.07 | 0.13 |
| RKV | 0.03 | 0.08 | 0.08 | 0.07 | 0.02 | 0.06 | 0.05 | 0.03 | 0.42 | 0.45 | 0.64 | 0.71 | 0.06 | 0.08 | 0.11 | 0.14 |
| SnapKV | **0.42** | **0.42** | **0.42** | 0.37 | **0.11** | **0.11** | **0.12** | **0.10** | **0.83** | **0.85** | 0.84 | **0.84** | **0.38** | **0.42** | **0.41** | **0.41** |
| StreamingLLM | 0.03 | 0.06 | 0.09 | 0.14 | 0.03 | 0.02 | 0.06 | 0.08 | 0.24 | 0.39 | 0.52 | 0.69 | 0.03 | 0.03 | 0.06 | 0.15 |

Table 4: **DeepSeek-R1-Distill-Llama-8B.** Performance using varying compressions and budgets.

| DeepSeek-R1-Distill-Llama-8B | GSM8K | | | | Math500 | | | | CSQA | | | | OBQA | | | |
|---|---|---|---|---|---|---|---|---|---|---|---|---|---|---|---|---|
| | 128 | 256 | 384 | 512 | 128 | 256 | 384 | 512 | 128 | 256 | 384 | 512 | 128 | 256 | 384 | 512 |
| Full | 0.70 | | | | 0.46 | | | | 0.75 | | | | 0.84 | | | |
| ShadowKV | 0.51 | | | | 0.34 | | | | 0.20 | | | | 0.31 | | | |
| H2O | 0.37 | 0.53 | 0.62 | 0.61 | 0.20 | 0.31 | 0.36 | 0.36 | 0.48 | 0.72 | 0.73 | **0.73** | 0.48 | 0.78 | **0.83** | **0.84** |
| Knorm | 0.00 | 0.09 | 0.19 | 0.28 | 0.00 | 0.01 | 0.02 | 0.06 | 0.05 | 0.28 | 0.54 | 0.66 | 0.03 | 0.27 | 0.57 | 0.70 |
| RKV | 0.05 | 0.04 | 0.14 | 0.17 | 0.03 | 0.05 | 0.02 | 0.02 | 0.07 | 0.11 | 0.16 | 0.35 | 0.07 | 0.07 | 0.19 | 0.32 |
| SnapKV | **0.72** | **0.72** | **0.74** | **0.72** | **0.42** | **0.44** | **0.41** | **0.41** | **0.74** | **0.73** | **0.74** | **0.73** | **0.82** | **0.83** | **0.83** | 0.81 |
| StreamingLLM | 0.06 | 0.25 | 0.39 | 0.56 | 0.03 | 0.09 | 0.21 | 0.29 | 0.04 | 0.14 | 0.35 | 0.50 | 0.07 | 0.15 | 0.32 | 0.52 |
| | ReClor | | | | DROP | | | | StrategyQA | | | | FOLIO | | | |
| | 128 | 256 | 384 | 512 | 128 | 256 | 384 | 512 | 128 | 256 | 384 | 512 | 128 | 256 | 384 | 512 |
| Full | 0.51 | | | | 0.14 | | | | 0.74 | | | | 0.47 | | | |
| ShadowKV | 0.27 | | | | 0.09 | | | | 0.80 | | | | 0.33 | | | |
| H2O | 0.03 | 0.08 | 0.23 | 0.38 | 0.06 | 0.07 | 0.10 | 0.11 | 0.25 | **0.69** | **0.77** | **0.79** | 0.07 | 0.37 | 0.41 | **0.46** |
| Knorm | 0.00 | 0.00 | 0.02 | 0.10 | 0.00 | 0.01 | 0.01 | 0.05 | 0.06 | 0.36 | 0.57 | 0.70 | 0.00 | 0.03 | 0.11 | 0.21 |
| RKV | 0.04 | 0.03 | 0.03 | 0.11 | 0.03 | 0.03 | 0.05 | 0.07 | 0.08 | 0.35 | 0.50 | 0.63 | 0.03 | 0.08 | 0.13 | 0.26 |
| SnapKV | **0.52** | **0.53** | **0.56** | **0.51** | **0.17** | **0.15** | **0.15** | **0.16** | **0.68** | 0.66 | 0.64 | 0.68 | **0.46** | **0.45** | **0.49** | **0.46** |
| StreamingLLM | 0.00 | 0.00 | 0.01 | 0.06 | 0.02 | 0.02 | 0.09 | 0.13 | 0.03 | 0.11 | 0.36 | 0.56 | 0.00 | 0.01 | 0.04 | 0.09 |

### 4.3 MAX TOKEN ABLATION

As explained in setup, we chose a max token length of 2048 both because we find that the mean token length over datasets is under this budget and to better assess performance in a compute-bound setting. However, we study the effect of max token limit on performance under a fixed budget of 1024 for MATH500 for R1-Distill-Qwen7B in Figure 7. We find that performance improves significantly for all methods initially, but then SnapKV-D overtakes all methods for all other max token limits.

### 4.4 LARGE MODEL COMPARISON

We determine whether our observed trends hold for a larger reasoning model, R1-Distill-Qwen-14B in Table 6. We examine the performance of all methods on the more challenging GSM8K and MATH500. Unsurprisingly, base accuracies do improve, but more importantly, we observe that again, the heavy-hitter methods H2O and SnapKV-D outperform their competitors by a significant margin indicating that larger reasoning models still benefit from attention-based eviction.

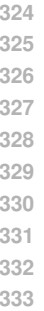

Figure 2: **Latency vs Budget.** Average generation time per token (ms) versus KV-cache budget for eviction strategies. KNorm and StreamingLLM speed up markedly with larger budgets, H2O improves more modestly, while SnapKV-D is slow at small budgets.

Table 5: **End-to-end token throughput (tokens/s) on GSM8K.** 50 sample average is reported.

| Method / Budget | 128 | 256 | 384 | 512 |
|---|---|---|---|---|
| Full | 30.71 | 30.71 | 30.71 | 30.71 |
| H2O | 25.81 | 23.41 | 23.93 | 25.16 |
| Knorm | 27.38 | 28.07 | 28.35 | 27.56 |
| R-KV | 29.10 | 27.72 | 28.55 | 30.04 |
| StreamingLLM | 27.04 | 27.55 | 27.12 | 29.39 |
| SnapKV | 27.46 | 25.57 | 26.24 | 27.60 |

Table 6: **R1-Distill-Qwen14B**. Cache budgets = [128, 256, 384, 512]. We examine the performance of various compression methods for a larger reasoning model. Winner per budget in bold.

| Method | GSM8K | | | | MATH500 | | | |
|---|---|---|---|---|---|---|---|---|
| | 128 | 256 | 384 | 512 | 128 | 256 | 384 | 512 |
| full | 0.81 | | | | 0.47 | | | |
| shadowkv | 0.53 | | | | 0.38 | | | |
| h2o | 0.33 | 0.56 | 0.62 | 0.64 | 0.20 | 0.27 | 0.31 | 0.31 |
| knorm | 0 | 0.02 | 0.08 | 0.21 | 0 | 0 | 0 | 0.02 |
| rkv | 0.02 | 0.05 | 0.16 | 0.30 | 0.00 | 0.00 | 0.03 | 0.09 |
| snapkv | **0.80** | **0.82** | **0.81** | **0.78** | **0.43** | **0.44** | **0.42** | **0.45** |
| streaming_llm | 0.07 | 0.27 | 0.50 | 0.59 | 0.02 | 0.17 | 0.26 | 0.35 |

## 4.5 CACHE BUDGET VS OUTPUT LENGTH

We study the effects of cache budget on output generation lengths in Figure 4. Fascinatingly, lower budgets are capable of triggering longer reasoning traces, revealing a hidden tradeoff between cache budget and inference costs specifically for reasoning models. KNorm, arguably the lowest performing strategy, tends to cause the greatest elongation of outputs. In Section A.2, we examine one such non-terminating output that demonstrates repetitive, dead-end chain-of-thought.

## 4.6 ATTENTION AS AN INDICATOR OF PERFORMANCE

All eviction methods tested propose to capture important tokens via ad-hoc strategies either explicitly or implicitly relating to attention: H2O examines at accumulated attention across the entire sequence, SnapKV examines attention with regards to an observation window, KNorm uses small key norms as a proxy for high-attention, StreamingLLM retains recent tokens and the sink (initial) to effectively approximate the attention distribution. We examine how much attention is actually lost through these various compression methods. For this study, we compare the absolute difference between the attention scores of each head pre- and post-eviction for GSM8K, which we refer to as attention loss following other recent literature (Liu et al., 2024a; Devoto et al., 2024). The trend is striking: in order

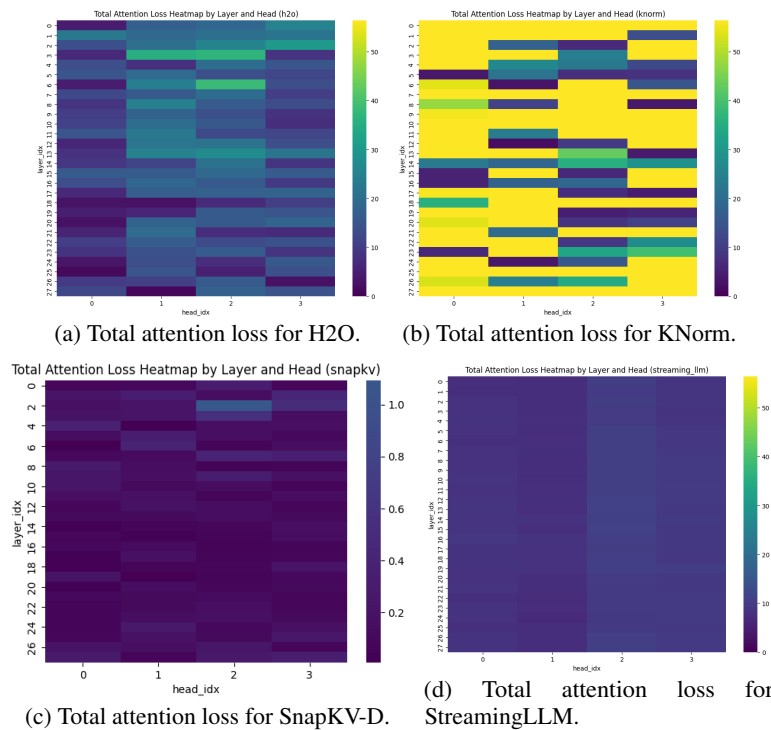

(a) Total attention loss for H2O.  (b) Total attention loss for KNorm.

(c) Total attention loss for SnapKV-D.  (d) Total attention loss for StreamingLLM.

Figure 3: **Attention Loss Heatmaps**. We visualize attention loss at every compression step for a question in GSM8K. The attention loss over each head is summed up over every layer. We observe that higher performance correlates with less attention loss.

of least to most attention loss: SnapKV-D, H2O, StreamingLLM, and KNorm. This correlates with average performance reported in Tables 2-4.

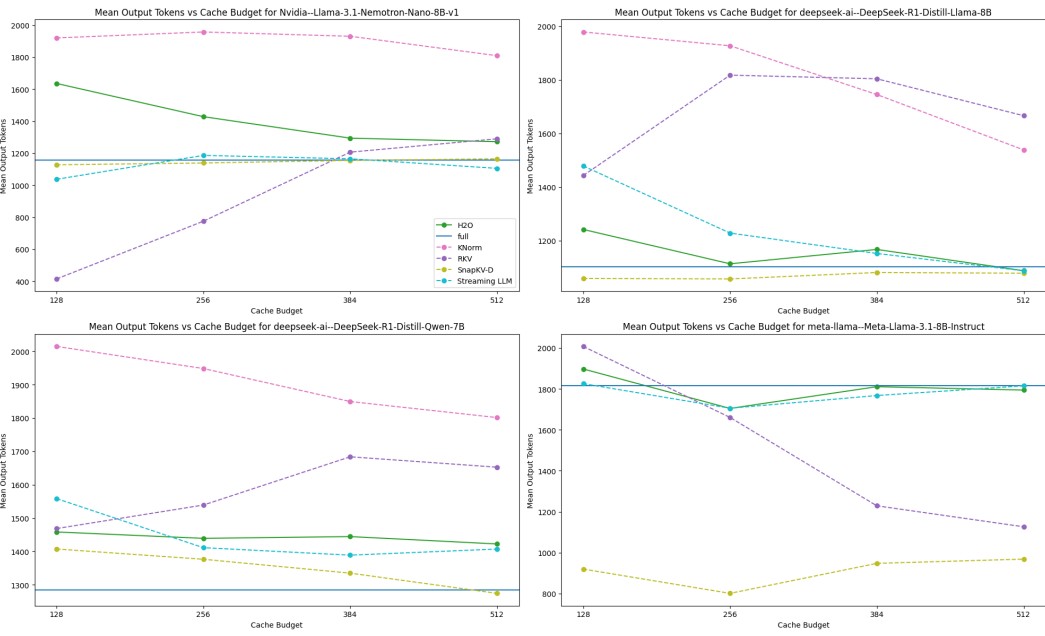

Figure 4: **Budget vs Output Length**. We observe that several compression methods, especially at lower budgets, ultimate produce longer outputs than the base full cache model.

## 4.7 ABLATION ANALYSIS & HIGH-LEVEL TRENDS

***Attention is the most versatile estimator for reasoning models.*** SnapKV-D and H2O are the most dominant, significantly outcompeting nearly all compression strategies across all budget constraints and datasets for our reasoning models. These methods rely on accumulated attention scores to determine the most important tokens to retain. (i.e., "heavy hitters"). While both maintain a recency window, H2O is focused on heavy hitters with regard to the current token, while SnapKV (and consequently, SnapKV-D) finds heavy hitters with respect to an observation window at the end of the current sequence. The latter approach is more effective, routinely defeating H2O. The observation window was previously known to work well for prompt compression, but not for long decoding.

To further verify the utility of heavy-hitters for reasoning, in Appendix A.4, we track how many critical keywords for reasoning in GSM8K questions are present in the cache for each method, across models and budgets. Table 8 demonstrates that SnapKV-D and H2O retain critical tokens at a higher rate than other methods. Since they are exclusively focused on extracting heavy-hitters, *this implies that critical keywords display high accumulated attention throughout reasoning.* Figure 5 shows that the density of keywords in decoded context is much higher for uncompressed reasoning models versus non-reasoning models, thus their maintenance in the cache is ostensibly crucial for computation.

***No singular strategy is dominant for the non-reasoning Llama-3.1-8B-Instruct.*** For models that do not produce reasoning traces, the optimal choice of strategy is dataset-dependent. For example, while StreamingLLM excels at GSM8K, it is less effective on all other task types. While SnapKV-D and H2O are capable of winning most settings for several datasets, other methods, such as KNorm and StreamingLLM, can win. According to Figure 5, keywords occur at a lower density in non-reasoning outputs, thus their maintenance might be less critical compared to reasoning outputs.

***Eviction lags full cache performance for reasoning models.*** According to Table 1, all compression strategies can defeat the full cache performance of Llama-3.1-8B-Instruct on at least one setting (with H2O and SnapKV-D frequently achieving this). However, for reasoning models, this trend only holds true for SnapKV-D. While H2O is still second best compared to other strategies, it significantly lags full cache performance on nearly every dataset. As noted in Figure 4, H2O results in significantly longer reasoning traces than SnapKV-D, which occasionally do not terminate.

***Cache compression can cost more computation.*** Interestingly, according to Figure 4, eviction strategies can result in more "talkative" reasoning models, generating noticeably longer sequences compared to the full cache setting, while this does not occur for Llama-3.1-8B-Instruct. In Section A.2, we show this phenomenon at work, where KNorm results in long circular babble for Deepseek-R1-Distill-Llama-8B that never produces an answer. At lower budgets, eviction occurs more frequently, resulting in a higher likelihood of critical token eviction, resulting in longer reasoning.

***Practical Guidance.*** Although heavy-hitter methods dominate, selection of other methods may still prove appropriate. We summarize key selection rules. (1) Regardless of method, avoid a micro-budget. Performance stabilizes rapidly and very small budgets can counterintuitively increase the length of the output. (2) For large budgets, $B > 1024$, StreamingLLM is superior with smaller max token limits according to Figure 7. For any other budget and max token limit, SnapKV-D and H2O are preferable. (3) Use a larger window size for SnapKV-D. This decreases the frequency of eviction and, consequently, computational overhead with minimal performance differences (Table 10). (4) For reasoning models, accumulated attention scores are a high-quality token importance metric, therefore, lead with heavy-hitter methods for compression.

## 5 CONCLUSION

In this work, we comprehensively assessed the performance of several popular KV cache compression strategies on reasoning tasks. For a non-reasoning model, we find that no singular method is dominant. However, for reasoning models, we demonstrate that attention-based eviction methods such as H2O and SnapKV-D perform extraordinarily well on a variety of reasoning tasks, even occasionally exceeding full cache performance. Furthermore, this generalizes to a larger model, R1-Distill-Qwen-14B. We also discover that it is possible, especially at lower budgets, for compression strategies to produce longer reasoning traces, thus revealing an under-considered tradeoff between memory and inference costs.

## 6 ETHICS STATEMENT

We do not anticipate any notable negative societal impacts stemming from this results discussed in this work. However, we do note that KV cache compression is capable of altering outputs and thus must be exercised with care in sensitive domains to ensure that content is not produced which significantly deviates from uncompressed models.

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

# A APPENDIX

## A.1 GENERATION LENGTHS

In Table 7, we report the mean generation lengths for all methods across and models on MATH-500, the dataset which elicits the longest responses. To keep these tables concise, we averaged output lengths over all budgets.

Table 7: Mean output tokens generated by different models under various strategies for Math500.

| Strategy | Model | Mean Output Tokens |
|---|---|---|
| full | Nvidia--Llama-3.1-Nemotron-Nano-8B-v1 | 1616.275 |
| full | deepseek-ai--DeepSeek-R1-Distill-Llama-8B | 1727.18 |
| full | deepseek-ai--DeepSeek-R1-Distill-Qwen-7B | 1728.84 |
| h2o | Nvidia--Llama-3.1-Nemotron-Nano-8B-v1 | 1753.4075 |
| h2o | deepseek-ai--DeepSeek-R1-Distill-Llama-8B | 1763.805 |
| h2o | deepseek-ai--DeepSeek-R1-Distill-Qwen-7B | 1767.7275 |
| knorm | Nvidia--Llama-3.1-Nemotron-Nano-8B-v1 | 1987.80875 |
| knorm | deepseek-ai--DeepSeek-R1-Distill-Llama-8B | 2001.07625 |
| knorm | deepseek-ai--DeepSeek-R1-Distill-Qwen-7B | 1967.4575 |
| snapkv | Nvidia--Llama-3.1-Nemotron-Nano-8B-v1 | 1667.895 |
| snapkv | deepseek-ai--DeepSeek-R1-Distill-Llama-8B | 1790.165 |
| snapkv | deepseek-ai--DeepSeek-R1-Distill-Qwen-7B | 1794.0275 |
| streaming_llm | Nvidia--Llama-3.1-Nemotron-Nano-8B-v1 | 1375.70375 |
| streaming_llm | deepseek-ai--DeepSeek-R1-Distill-Llama-8B | 1655.0375 |
| streaming_llm | deepseek-ai--DeepSeek-R1-Distill-Qwen-7B | 1698.25625 |

## A.2 A VERBOSE, WRONG ANSWER

Below, we present a verbose, incorrect answer generated for Deepseek-R1-Distill-Llama-8B via KNorm on a budget of 256 tokens.

```
Define:

- Let a be the value of a small button (SB).
- Then, 1 SS = a
- 1 large button (LB) is worth 3 small buttons, so LB = 3a
- Alternatively, 1 LB = 3a
- 1 LS can be converted to 1 LB or 3 SS, so 1 LS = 1 LB or 3*(1a)? Wait,
    small stickers themselves are worth 1a.

Hold on, perhaps clarifying units.

Wait, sorry, the problem states:

"Lorraine has ... stickers and large buttons..."

Wait, maybe it's better to model it with equations.

Let me denote:

- Value of 1 small sticker = x
- Then, 1 medium sticker would be higher, but in our case, 1 large
    sticker (LS) is worth either 1 large button (LB) or 3 small buttons (
    SB).

But we don't know whether a small button is same as a small sticker? Let'
    s see.

The problem says:
```

Table 8: Strategy vs. Critical Token Retention Rate. Cache budgets (MB) = [128, 256, 384, 512]. Models: Llama-3.1-8B-Instruct (ML), DeepSeek-R1-Distill-Qwen-7B (DQ), Nemotron-Nano-8B-v1 (LN), DeepSeek-R1-Distill-Llama-8B (DL).

| Benchmark | ML | | | | DQ | | | | LN | | | | DL | | | |
| --- | --- | --- | --- | --- | --- | --- | --- | --- | --- | --- | --- | --- | --- | --- | --- | --- |
| | 128 | 256 | 384 | 512 | 128 | 256 | 384 | 512 | 128 | 256 | 384 | 512 | 128 | 256 | 384 | 512 |
| GSM8K full | 77.84 | | | | 77.46% | | | | 81.62% | | | | 75.59% | | | |
| h2o | 68.11% | 68.11% | 69.19% | 70.27% | 72.30% | 72.30% | 73.24% | 74.18% | 68.11% | 68.11% | 69.19% | 70.27% | 71.61% | 71.61% | 72.30% | 72.30% |
| knorm | 68.11% | 68.11% | 69.19% | 69.73% | 67.60% | 67.60% | 68.72% | 69.27% | 65.48% | 65.48% | 66.67% | 67.26% | 68.11% | 68.11% | 69.19% | 69.73% |
| SnapKV | 68.11% | 68.11% | 68.65% | 69.73% | 72.30% | 72.30% | 73.24% | 74.18% | 68.11% | 68.11% | 69.19% | 70.27% | 68.11% | 68.11% | 69.19% | 70.27% |
| streaming_llm | 67.03% | 67.03% | 67.57% | 69.73% | 71.36% | 71.36% | 72.30% | 73.24% | 67.03% | 67.03% | 68.11% | 69.19% | 67.03% | 67.03% | 68.11% | 69.19% |

```
- A large button is worth three small stickers. So 1 LB = 3 SS.

So, 1 SS = 1/3 LB.

Or, alternatively, 1 LB = 3 SS.

So, yeah, 1 LB = 3 SS.

On the flip side, 1 LS can be either 1 LB or 3 SS.

But note that 1 LB is equal to 3 SS.

So, the question becomes:

Traders use sticker-based currencies.

Given that, let me
```

## A.3    LLM USAGE

We used LLMs to stylize tables and figures. We also used LLMs to polish grammar but not to produce any writing itself.

## A.4    RETENTION OF CRITICAL TOKENS

To understand why reasoning oriented models such as DeepSeek-R1-Distill-Qwen-7B and DeepSeek-R1-Distill-Llama-8B exhibit different performance patterns, we evaluate GSM8K questions and measure how well critical tokens (e.g., names, quantities, and key entities) are retained in the KV cache at the final decoding step. Table 8 reports the critical token retention rate, defined as the fraction of critical tokens that remain available in the cache at the end of generation. Figure 5 further visualizes a randomly selected GSM8K example, highlighting which critical tokens persist in the cache and how they appear in the model's final answer. From both the quantitative results and the visualization, we observe that reasoning models consistently retain a larger proportion of critical tokens than standard instruction tuned baselines. In particular, reasoning models tend to preserve task relevant concepts such as minutes, day, and similar numerical or semantic anchors, suggesting that they rely more directly on these tokens throughout the multistep reasoning process.

We find that reasoning-trained models consistently retain a larger proportion of critical tokens compared with standard instruction-tuned models. Both the numerical metrics and qualitative visualizations indicate that reasoning models rely more heavily on these critical tokens throughout the problem-solving process. In the example visualization, the reasoning model generates more critical tokens in its final answer, which helps explain why heavy-hitter or attention-based token-retention strategies are particularly effective for reasoning models: critical tokens tend to persistently exhibit high-attention throughout decoding.

This observation directly supports our main conclusion that accumulated attention is the most effective importance heuristic for reasoning eviction. Since RL-distilled reasoning models naturally focus on preserving critical tokens, methods such as **H2O** and **SnapKV-D** yield larger performance gains by ensuring those tokens remain in the KV cache. In contrast, non-reasoning models do not exhibit this

DeepSeek-Distill-Qwen-7B

Okay, so I need to figure out how many minutes Kim slept on her bed that day . Let me start by understanding all the information given.

Firstly, normally Kim goes to sleep at 10 p.m. (which is 2 2 :00) and wakes up at 6 a.m. next morning (which would be 6 :00). So let's calculate her usual total sleeping time first before considering to day 's anomaly because sometimes it helps have some context.

From 10 p.m. to midnight is 2 hours, which is 1 2 0 minutes. Then from midnight to 6 a.m. is another 6 hours, which is 3 6 0 minutes.

So altogether, normal sleeping duration is 1 2 0 + 3 6 0 = 480 minutes each night.

But one day , something different happened. On this particular day :

- She went to sleep as usual? Or maybe not... Wait, no—the problem says "that day " meaning only that specific instance where we're analyzing what happened differently.

Wait—let me read again carefully:

"She installed a camera ... Kim usually goes to sleep at 10 p.m., but that day , she found herself waking up 5 minutes earlier than usual." Hmm...

Meta-Llama3-8B-Instruct

\## Step 1: Determine normal bedtime and wake-up time

Normal bedtime for Kim is 10 p.m., and normal wake-up time is 6 a.m.

\## Step 2 : Calculate total number of minutes slept normally per night

There are 8 hours between 10 p.m. and 6 a.m. Since there are 6 0 minutes in an hour, this means that under normal circumstances, Kim sleeps for (8 * 6 0) = 480 minutes every night.

\## Step 3: Find out how early Kim went to sleep that particular evening.

Since Kim fell asleep while sleepwalking from 2:15 to 2:38 AM but actually started waking up 5 minutes before regular wake-up time, we need to determine what time it would be if she had gone to bed at the normal time instead of falling asleep during the middle of the night.

\## Step 4: Convert wake-up times into just minutes past midnight

For calculation purposes, let's convert all relevant times into minutes since midnight:

- Normal wake-up time is 6 a.m., which equals 3 6 0 minutes past midnight (since 1 2 am = 0).

- Sleep period due to being awake early: From 2:15 A.M. till 2:38 A.M. is ( 2 3 *6 0)+ 15 =1 38 9+ 15 =1404 minutes past midnight

Figure 5: **A generated answer by DeepSeek-R1-Distill-Qwen-7B and Meta-Llama3-8B-Instruct.** Critical tokens rendered in blue and red boxes respectively. Reasoning models outputs contain a higher density of critical tokens. `Prompt`: *Kim sleepwalks, to monitor her sleeping hours, she installs a camera in her room. Kim usually goes to sleep at 10 p.m. and wakes up at 6 a.m. She is a sleepwalker. One day, after reviewing the cameras, she finds that she woke up that day and was sleepwalking from 2:15 to 2:38 am. Also, that day she woke up 5 minutes earlier than usual to go to the bathroom. How many minutes did she sleep on her bed that day?*

strong reliance on critical tokens, so no single cache-retention strategy is universally optimal across model types.

### A.5 MEMORY PERFORMANCE ANALYSIS

In this section, we report the peak memory usage between methods.

Table 9: Peak allocated GPU memory (GB) by press and cache budget on GSM8K.

| press | 128 | 256 | 384 | 512 |
|---|---|---|---|---|
| Full | 14.88 | 14.88 | 14.88 | 14.88 |
| H2O | 14.81 | 14.81 | 14.82 | 14.83 |
| KNorm | 14.79 | 14.80 | 14.81 | 14.83 |
| R-KV | 14.80 | 14.81 | 14.83 | 14.84 |
| StreamingLLM | 14.79 | 14.80 | 14.81 | 14.83 |

### A.6 WINDOW SIZE

SnapKV-D using a sliding window of fixed size $w$ to determine critical tokens. In particular, after the budget is filled to capacity, every $w$ decoding steps, SnapKV-D measures the attention scores of all current tokens in the cache against the $w$ most recent tokens (the aggregation is described in (Li et al., 2024)). Those with the lowest scores are dropped to meet the budget again.

In our core experiments, we set $w = 128$, which is recommended by both the authors and the kvpress library. We perform an ablation size over window size in Table 10 The effects are only noticeable at lower budgets, where smaller window size forces more frequent re-assessment of critical tokens in the cache, thus maintaining tokens more relevant to incoming context. However, this costs more overhead: if the total decoded output is length $N$, we are performing $N/w$ applications of SnapKV-D.

Table 10: Model Performance across Cache Budgets and Window Sizes for SnapKV

| Model | Window Size | Cache Budget | | | |
|---|---|---|---|---|---|
| | | 128 | 256 | 384 | 512 |
| deepseek-ai/DeepSeek-R1-Distill-Llama-8B | 16 | 0.71 | 0.69 | 0.80 | 0.69 |
| | 32 | 0.74 | 0.71 | 0.73 | 0.73 |
| | 64 | 0.72 | 0.72 | 0.74 | 0.72 |
| | 128 | 0.72 | 0.69 | 0.75 | 0.70 |
| Nvidia/Llama-3.1-Nemotron-Nano-8B-v1 | 16 | 0.71 | 0.69 | 0.66 | 0.66 |
| | 32 | 0.71 | 0.66 | 0.68 | 0.67 |
| | 64 | 0.65 | 0.63 | 0.66 | 0.66 |
| | 128 | 0.72 | 0.73 | 0.72 | 0.67 |
| deepseek-ai/DeepSeek-R1-Distill-Qwen-7B | 16 | 0.74 | 0.68 | 0.65 | 0.67 |
| | 32 | 0.69 | 0.71 | 0.68 | 0.68 |
| | 64 | 0.67 | 0.67 | 0.70 | 0.71 |
| | 128 | 0.66 | 0.65 | 0.70 | 0.72 |

Our results illustrate that for larger budgets, wider window sizes should be used since this both improves accuracy and reduces computation.

## A.7 PYRAMIDKV ANALYSIS

PyramidKV is a dynamic KV-cache compression method that is built around the idea of pyramidal information funneling: in early Transformer layers, attention is spread broadly over many tokens, while in deeper layers it becomes concentrated on a small subset of salient tokens. Under a fixed overall KV budget, PyramidKV therefore allocates larger cache sizes to lower layers and progressively smaller caches to higher layers, forming a pyramid-shaped retention profile across depth. Within each layer, it uses attention patterns to decide which keys and values to keep (e.g., tokens that are strongly attended to by query/instruction tokens are preferentially retained), so that the cache focuses on the most informative context while still substantially reducing memory usage.

| Dataset | Model | Budget | | | |
|---|---|---|---|---|---|
| | | 128 | 256 | 384 | 512 |
| **GSM8K** | | | | | |
| gsm8k | DeepSeek-R1-Distill-Llama-8B | 0.01 | 0.01 | 0.10 | 0.22 |
| gsm8k | DeepSeek-R1-Distill-Qwen-7B | 0.00 | 0.03 | 0.09 | 0.25 |
| gsm8k | Meta-Llama-3.1-8B-Instruct | 0.03 | 0.38 | 0.72 | 0.79 |
| gsm8k | Llama-3.1-Nemotron-Nano-8B-v1 | 0.01 | 0.01 | 0.03 | 0.20 |
| **MATH500** | | | | | |
| math500 | DeepSeek-R1-Distill-Llama-8B | 0.00 | 0.00 | 0.00 | 0.00 |
| math500 | DeepSeek-R1-Distill-Qwen-7B | 0.00 | 0.01 | 0.01 | 0.06 |
| math500 | Meta-Llama-3.1-8B-Instruct | 0.01 | 0.05 | 0.16 | 0.25 |
| math500 | Llama-3.1-Nemotron-Nano-8B-v1 | 0.02 | 0.02 | 0.04 | 0.04 |

Table 11: Test accuracy on the GSM8K and MATH500 test sets for each model and KV-cache budget. Budgets (128, 256, 384, 512) index the maximum KV-cache size in tokens, and each cell reports the corresponding accuracy at that budget.

## A.8 COMPARISON WITH A SPARSE ATTENTION METHOD

Although our main benchmark evaluates KV Cache pressing methods, we extend our study to compare these results with other architectural categories, such as sparse attention methods. Such methods (Gao et al., 2025; Yuan et al., 2025) train their architectures to enforce sparse attention computations by learning to identify and cluster critical tokens. This is in contrast to eviction methods which are generally training-free. Furthermore, these methods are not memory-bound and host the full KV cache.

We perform a comparative evaluation of sparse decoding modeling with SeerAttention using the `SeerAttention-Decode-R1-Distill-Qwen-14B` model and present results in Table 12. This analysis is performed on the GSM8K benchmark using a randomly sampled subset of 100 questions. We can see that SnapKV-D and SeerAttention are close in performance with SnapKV-D as slightly better. *Further note that SeerAttention must maintain the full cache which scales with sequence length, while SnapKV-D maintains a fixed size cache.*

Table 12: **SeerAttention-R1-Distill-Qwen14B**. Cache budgets = [128, 256, 384, 512]. We examine the performance of SeerAttention together with H2O and SnapKV-D. Winner per budget in bold.

| Method | GSM8K | | | |
|---|---|---|---|---|
| | 128 | 256 | 384 | 512 |
| H2O | 0.33 | 0.56 | 0.62 | 0.64 |
| SnapKV-D | **0.80** | **0.82** | 0.81 | **0.78** |
| SeerAttention | 0.66 | 0.80 | **0.82** | 0.70 |

### A.9   COMPUTATIONAL OVERHEAD

In this section, we describe the asymptotic computational overhead of each method. More specifically, in Table 13, we report the computational complexity of cache eviction throughout the decoded sequence. Once the cache is evicted down to the budget $B$, the attention calculation is $\mathcal{O}_d(1)$ (since there are only $B$ tokens of dimension $d$ to compute attention over). Thus, we are interested in comparing the complexity of *evaluating token importance itself.*

Any method relying on accumulated attention scores (H2O & SnapKV-D) incurs a greater cost. StreamingLLM and KNorm are comparatively cheap; the former just keeps a few sink tokens and recent context, while the latter simply evicts the token with the largest key norm. These latencies are reflected accordingly in Figure2.

Table 13: Decoding computational overhead of importance estimation for our tested KV cache compression methods. $B$ is the cache budget, $N$ is the decoded sequence length, and $d$ the key dimension. For simplicity, we are assuming that the budget is filled after pre-fill and that each attention layer is single-head.

| Method | Overhead | Approach |
|---|---|---|
| StreamingLLM | $\mathcal{O}(1)$ | Maintain sink token + recent tokens. |
| H2O | $\mathcal{O}(NBd)$ | Next token in, token in cache with lowest average accumulated attention score is out. |
| SnapKV-D | $\mathcal{O}(\frac{N}{w}Bd)$ | Keep the next window of $w$ tokens, evict tokens in the cache with low accumulated attention score against the window. |
| R-KV | $\mathcal{O}(\frac{N}{w}B^2d)$ | Measures token redundancy (key cache self-product) and accumulated attention scores against the query every $w$ steps. |
| Knorm | $\mathcal{O}(N)$ | Evicts the token in the cache with the largest $\ell_2$ norm. |

### A.10   THE TRANSFORMER DECODER ARCHITECTURE AND INFERENCE

We visualize decoder-only inference and the role of the cache. Figure 6 (a) exhibits the $Q$, $K$, and $V$ vectors along with the self-attention mechanism. Figure 6 (b) demonstrate the decoding KV cache bottleneck on memory.

### A.11   HYPERPARAMETER DETAILS

In this section, we describe the hyperparameter details for all eviction methods. We tend towards the default hyper-parameters set by KVPress, which are typically author-recommend selections.

**H2O:**   $H_2$ tokens kept

**PyramidKV:**   window size 64, kernel size 5, $\beta = 20$

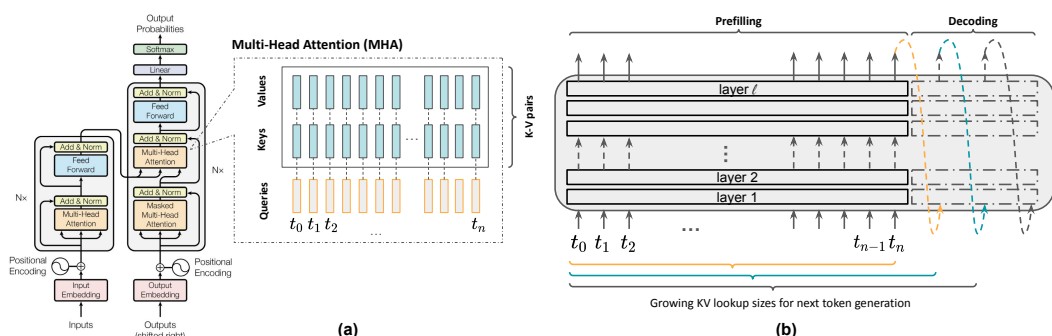

Figure 6: **Overview of the Transformer Decoder Architecture and the Inference Bottleneck.** (a) The standard Transformer decoder architecture (left) and the Multi-Head Attention (MHA) mechanism (right). In MHA, Query vectors representing the current context attend to a sequence of Key-Value (K-V) pairs from all previous tokens. Such K-V pairs form the basis of the KV cache. (b) The two-phase inference process in autoregressive generation. During Prefilling, the tokens in the input context are processed in parallel to populate the initial KV cache across all layers. During Decoding, each new token is generated sequentially. This requires recomputing the entire set of the preceding KV entries at each step, causing the lookup size to grow linearly with the sequence length.

**StreamingLLM:** Sink retention, first 4 tokens.

**SnapKV-D:** Observation window size 128.

**KNorm:** $k = 2$

**R-KV:** window size 8, buffer interval 128, kernel size 5.

### A.12 MAX TOKEN ABLATION

We study the effects of max token limit on performance under a fixed budget of 1024 tokens. Results are presented in Figure 7.

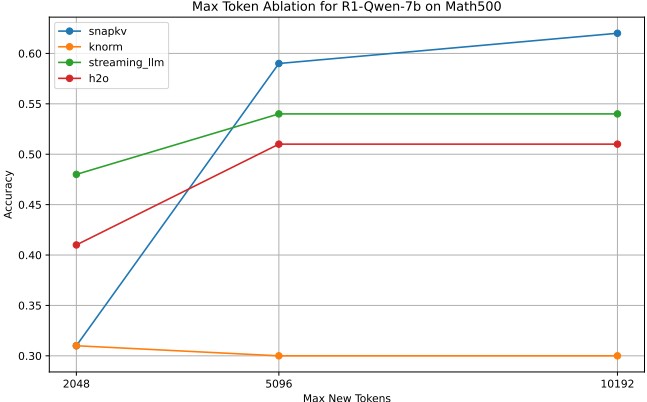

Figure 7: **Performance versus max tokens permitted.** Fixed budget of 1024.

### A.13 CONFIDENCE INTERVALS

For each configuration (model, dataset, cache budget, method), we estimate uncertainty by computing a binomial confidence interval over correctness across runs, as shown in Table 14. Specifically, we evaluate 3 independent seeds, each on 100 questions, and treat the resulting 300 binary outcomes (correct/incorrect) as Bernoulli trials with unknown success probability. The sample accuracy for that configuration is the proportion of correct answers over these 300 trials, and we then compute a 95% Wilson score confidence interval for this underlying accuracy parameter (using the normal

approximation with $z = 1.96$. The table cells in this appendix report only these Wilson intervals [L,U], omitting the point estimates, which are shown separately in the main results Tables 1-4.

Table 14: Confidence intervals across cache budgets (Continued)

| Budget | 128 | 256 | 384 | 512 | 128 | 256 | 384 | 512 |
|---|---|---|---|---|---|---|---|---|
| **Llama-3.1-8B-Instruct** | | | | | | | | |
| **Method** | | **GSM8K** | | | | **Math500** | | |
| Full | | [0.838, 0.912] | | | | [0.337, 0.446] | | |
| ShadowKV | | [0.270, 0.375] | | | | [0.177, 0.270] | | |
| H2O | **[0.574, 0.683]** | **[0.719, 0.814]** | [0.773, 0.859] | [0.783, 0.868] | [0.251, 0.354] | [0.279, 0.385] | [0.279, 0.385] | **[0.308, 0.416]** |
| Knorm | [0.031, 0.081] | [0.473, 0.586] | [0.677, 0.777] | [0.773, 0.859] | [0.016, 0.056] | [0.141, 0.227] | [0.177, 0.270] | [0.279, 0.385] |
| RKV | [0.088, 0.162] | [0.289, 0.395] | [0.444, 0.556] | [0.434, 0.546] | [0.016, 0.056] | [0.071, 0.139] | [0.123, 0.206] | [0.159, 0.249] |
| SnapKV | [0.473, 0.586] | [0.493, 0.605] | [0.503, 0.615] | [0.473, 0.586] | [0.159, 0.249] | [0.168, 0.260] | [0.150, 0.238] | [0.159, 0.249] |
| StreamingLLM | [0.214, 0.312] | [0.698, 0.796] | **[0.794, 0.877]** | **[0.827, 0.903]** | [0.079, 0.150] | [0.214, 0.312] | [0.270, 0.375] | [0.298, 0.406] |
| **Method** | | **CSQA** | | | | **OBQA** | | |
| Full | | [0.719, 0.814] | | | | [0.794, 0.877] | | |
| ShadowKV | | [0.159, 0.249] | | | | [0.260, 0.364] | | |
| H2O | **[0.688, 0.786]** | [0.709, 0.805] | **[0.719, 0.814]** | **[0.719, 0.814]** | [0.783, 0.868] | **[0.816, 0.895]** | **[0.816, 0.895]** | **[0.816, 0.895]** |
| Knorm | [0.289, 0.395] | **[0.719, 0.814]** | [0.698, 0.796] | [0.709, 0.805] | [0.356, 0.466] | [0.740, 0.832] | [0.794, 0.877] | [0.773, 0.859] |
| RKV | [0.308, 0.416] | [0.564, 0.673] | [0.709, 0.805] | [0.719, 0.814] | [0.177, 0.270] | [0.605, 0.711] | [0.719, 0.814] | [0.794, 0.877] |
| SnapKV | [0.646, 0.749] | [0.584, 0.692] | [0.656, 0.758] | [0.667, 0.768] | [0.677, 0.777] | [0.719, 0.814] | [0.667, 0.768] | [0.709, 0.805] |
| StreamingLLM | [0.159, 0.249] | [0.698, 0.796] | [0.709, 0.805] | **[0.719, 0.814]** | [0.105, 0.184] | [0.667, 0.768] | [0.794, 0.877] | [0.794, 0.877] |
| **Method** | | **ReClor** | | | | **DROP** | | |
| Full | | [0.544, 0.654] | | | | [0.114, 0.195] | | |
| ShadowKV | | [0.223, 0.323] | | | | [0.232, 0.333] | | |
| H2O | [0.270, 0.375] | [0.503, 0.615] | **[0.544, 0.654]** | [0.523, 0.634] | [0.088, 0.162] | **[0.105, 0.184]** | **[0.132, 0.217]** | **[0.132, 0.217]** |
| Knorm | [0.003, 0.029] | [0.150, 0.238] | [0.404, 0.517] | **[0.534, 0.644]** | [0.003, 0.029] | [0.054, 0.116] | [0.097, 0.173] | [0.097, 0.173] |
| RKV | [0.023, 0.069] | [0.168, 0.260] | [0.346, 0.456] | [0.483, 0.596] | [0.038, 0.093] | [0.046, 0.105] | [0.105, 0.184] | [0.079, 0.150] |
| SnapKV | **[0.473, 0.586]** | **[0.513, 0.625]** | [0.523, 0.634] | [0.493, 0.605] | **[0.114, 0.195]** | [0.088, 0.162] | [0.079, 0.150] | [0.088, 0.162] |
| StreamingLLM | [0.031, 0.081] | [0.168, 0.260] | [0.534, 0.644] | [0.523, 0.634] | [0.063, 0.128] | [0.079, 0.150] | [0.114, 0.195] | [0.123, 0.206] |
| **Method** | | **StrategyQA** | | | | **FOLIO** | | |
| Full | | [0.783, 0.868] | | | | [0.454, 0.566] | | |
| ShadowKV | | [0.625, 0.730] | | | | [0.279, 0.385] | | |
| H2O | **[0.762, 0.850]** | **[0.827, 0.903]** | [0.838, 0.912] | **[0.850, 0.921]** | [0.177, 0.270] | **[0.375, 0.487]** | [0.356, 0.466] | [0.375, 0.487] |
| Knorm | [0.414, 0.527] | [0.805, 0.886] | [0.838, 0.912] | [0.827, 0.903] | [0.009, 0.043] | [0.232, 0.333] | [0.337, 0.446] | [0.327, 0.436] |
| RKV | [0.544, 0.654] | [0.740, 0.832] | [0.719, 0.814] | [0.740, 0.832] | [0.046, 0.105] | [0.308, 0.416] | [0.385, 0.497] | [0.289, 0.395] |
| SnapKV | [0.730, 0.823] | [0.730, 0.823] | [0.762, 0.850] | [0.709, 0.805] | **[0.385, 0.497]** | [0.346, 0.456] | **[0.395, 0.507]** | **[0.404, 0.517]** |
| StreamingLLM | [0.079, 0.150] | [0.709, 0.805] | **[0.850, 0.921]** | [0.805, 0.886] | [0.016, 0.056] | [0.063, 0.128] | [0.204, 0.302] | [0.298, 0.406] |
| **Deepseek-R1-Distill-Qwen-7B** | | | | | | | | |
| **Method** | | **GSM8K** | | | | **Math500** | | |
| Full | | [0.646, 0.749] | | | | [0.414, 0.527] | | |
| ShadowKV | | [0.414, 0.527] | | | | [0.279, 0.385] | | |
| H2O | [0.168, 0.260] | [0.385, 0.497] | [0.454, 0.566] | [0.464, 0.576] | [0.105, 0.184] | [0.168, 0.260] | [0.242, 0.344] | [0.260, 0.364] |
| Knorm | [0.000, 0.013] | [0.000, 0.013] | [0.054, 0.116] | [0.123, 0.206] | [0.000, 0.013] | [0.003, 0.029] | [0.016, 0.056] | [0.031, 0.081] |
| RKV | [0.023, 0.069] | [0.046, 0.105] | [0.141, 0.227] | [0.251, 0.354] | [0.023, 0.069] | [0.023, 0.069] | [0.031, 0.081] | [0.132, 0.217] |
| SnapKV | **[0.615, 0.721]** | **[0.615, 0.721]** | **[0.646, 0.749]** | **[0.656, 0.758]** | **[0.327, 0.436]** | **[0.308, 0.416]** | **[0.308, 0.416]** | **[0.270, 0.375]** |
| StreamingLLM | [0.009, 0.043] | [0.150, 0.238] | [0.270, 0.375] | [0.385, 0.497] | [0.016, 0.056] | [0.088, 0.162] | [0.150, 0.238] | [0.214, 0.312] |
| **Method** | | **CSQA** | | | | **OBQA** | | |
| Full | | [0.615, 0.721] | | | | [0.730, 0.823] | | |
| ShadowKV | | [0.159, 0.249] | | | | [0.260, 0.364] | | |
| H2O | [0.385, 0.497] | [0.554, 0.663] | **[0.544, 0.654]** | **[0.584, 0.692]** | [0.366, 0.477] | [0.584, 0.692] | **[0.636, 0.740]** | [0.615, 0.721] |
| Knorm | [0.031, 0.081] | [0.097, 0.173] | [0.251, 0.354] | [0.366, 0.477] | [0.016, 0.056] | [0.031, 0.081] | [0.186, 0.281] | [0.327, 0.436] |
| RKV | [0.071, 0.139] | [0.063, 0.128] | [0.223, 0.323] | [0.289, 0.395] | [0.071, 0.139] | [0.071, 0.139] | [0.168, 0.260] | [0.214, 0.312] |
| SnapKV | **[0.594, 0.702]** | **[0.564, 0.673]** | [0.534, 0.644] | [0.554, 0.663] | **[0.656, 0.758]** | **[0.698, 0.796]** | [0.625, 0.730] | **[0.709, 0.805]** |
| StreamingLLM | [0.054, 0.116] | [0.105, 0.184] | [0.260, 0.364] | [0.424, 0.536] | [0.009, 0.043] | [0.079, 0.150] | [0.232, 0.333] | [0.317, 0.426] |
| **Method** | | **ReClor** | | | | **DROP** | | |
| Full | | [0.395, 0.507] | | | | [0.123, 0.206] | | |
| ShadowKV | | [0.223, 0.323] | | | | [0.105, 0.184] | | |
| H2O | [0.003, 0.029] | [0.023, 0.069] | [0.141, 0.227] | [0.232, 0.333] | [0.023, 0.069] | [0.046, 0.105] | [0.071, 0.139] | [0.071, 0.139] |
| Knorm | [0.000, 0.013] | [0.000, 0.013] | [0.003, 0.029] | [0.003, 0.029] | [0.000, 0.013] | [0.003, 0.029] | [0.003, 0.029] | [0.016, 0.056] |
| RKV | [0.023, 0.069] | [0.016, 0.056] | [0.009, 0.043] | [0.003, 0.029] | [0.023, 0.069] | [0.023, 0.069] | [0.016, 0.056] | [0.023, 0.069] |
| SnapKV | **[0.395, 0.507]** | **[0.337, 0.446]** | **[0.346, 0.456]** | **[0.375, 0.487]** | **[0.097, 0.173]** | [0.079, 0.150] | [0.088, 0.162] | [0.123, 0.206] |
| StreamingLLM | [0.000, 0.013] | [0.003, 0.029] | [0.003, 0.029] | [0.023, 0.069] | [0.023, 0.069] | [0.031, 0.081] | [0.054, 0.116] | [0.097, 0.173] |

*Continued on next page...*

Table 14: Confidence intervals (continued)...

| Budget | 128 | 256 | 384 | 512 | 128 | 256 | 384 | 512 |
|---|---|---|---|---|---|---|---|---|
| **Method** | **StrategyQA** | | | | **FOLIO** | | | |
| Full | [0.615, 0.721] | | | | [0.308, 0.416] | | | |
| ShadowKV | [0.544, 0.654] | | | | [0.279, 0.385] | | | |
| H2O | [0.279, 0.385] | **[0.584, 0.692]** | **[0.688, 0.786]** | **[0.667, 0.768]** | [0.016, 0.056] | [0.168, 0.260] | [0.186, 0.281] | [0.186, 0.281] |
| Knorm | [0.000, 0.013] | [0.088, 0.162] | [0.385, 0.497] | [0.534, 0.644] | [0.000, 0.013] | [0.003, 0.029] | [0.016, 0.056] | [0.031, 0.081] |
| RKV | [0.031, 0.081] | [0.105, 0.184] | [0.289, 0.395] | [0.366, 0.477] | [0.023, 0.069] | [0.016, 0.056] | [0.009, 0.043] | [0.038, 0.093] |
| SnapKV | **[0.544, 0.654]** | [0.534, 0.644] | [0.513, 0.625] | [0.574, 0.683] | **[0.251, 0.354]** | **[0.204, 0.302]** | **[0.260, 0.364]** | **[0.242, 0.344]** |
| StreamingLLM | [0.000, 0.013] | [0.031, 0.081] | [0.177, 0.270] | [0.366, 0.477] | [0.000, 0.013] | [0.003, 0.029] | [0.009, 0.043] | [0.016, 0.056] |
| **Nemotron-Nano-8B** | | | | | | | | |
| **Method** | **GSM8K** | | | | **Math500** | | | |
| Full | [0.584, 0.692] | | | | [0.395, 0.507] | | | |
| ShadowKV | [0.385, 0.497] | | | | [0.232, 0.333] | | | |
| H2O | [0.177, 0.270] | [0.395, 0.507] | [0.464, 0.576] | [0.513, 0.625] | [0.123, 0.206] | [0.195, 0.291] | [0.260, 0.364] | [0.279, 0.385] |
| Knorm | [0.003, 0.029] | [0.009, 0.043] | [0.063, 0.128] | [0.141, 0.227] | [0.003, 0.029] | [0.003, 0.029] | [0.016, 0.056] | [0.038, 0.093] |
| RKV | [0.023, 0.069] | [0.016, 0.056] | [0.063, 0.128] | [0.114, 0.195] | [0.009, 0.043] | [0.023, 0.069] | [0.016, 0.056] | [0.038, 0.093] |
| SnapKV | **[0.594, 0.702]** | **[0.574, 0.683]** | **[0.605, 0.711]** | **[0.605, 0.711]** | **[0.356, 0.466]** | **[0.385, 0.497]** | **[0.395, 0.507]** | **[0.375, 0.487]** |
| StreamingLLM | [0.016, 0.056] | [0.159, 0.249] | [0.346, 0.456] | [0.473, 0.586] | [0.009, 0.043] | [0.097, 0.173] | [0.177, 0.270] | [0.289, 0.395] |
| **Method** | **CSQA** | | | | **OBQA** | | | |
| Full | [0.454, 0.566] | | | | [0.584, 0.692] | | | |
| ShadowKV | [0.159, 0.249] | | | | [0.260, 0.364] | | | |
| H2O | [0.414, 0.527] | [0.434, 0.546] | **[0.464, 0.576]** | [0.454, 0.566] | [0.534, 0.644] | [0.534, 0.644] | [0.523, 0.634] | [0.564, 0.673] |
| Knorm | [0.308, 0.416] | [0.346, 0.456] | [0.385, 0.497] | [0.404, 0.517] | [0.270, 0.375] | [0.385, 0.497] | [0.424, 0.536] | [0.513, 0.625] |
| RKV | [0.232, 0.333] | [0.251, 0.354] | [0.366, 0.477] | [0.356, 0.466] | [0.298, 0.406] | [0.385, 0.497] | [0.454, 0.566] | [0.454, 0.566] |
| SnapKV | **[0.434, 0.546]** | **[0.444, 0.556]** | [0.454, 0.566] | **[0.473, 0.586]** | **[0.625, 0.730]** | **[0.574, 0.683]** | **[0.605, 0.711]** | **[0.605, 0.711]** |
| StreamingLLM | [0.308, 0.416] | [0.385, 0.497] | [0.404, 0.517] | [0.444, 0.556] | [0.308, 0.416] | [0.404, 0.517] | [0.464, 0.576] | [0.564, 0.673] |
| **Method** | **ReClor** | | | | **DROP** | | | |
| Full | [0.424, 0.536] | | | | [0.079, 0.150] | | | |
| ShadowKV | [0.223, 0.323] | | | | [0.079, 0.150] | | | |
| H2O | [0.159, 0.249] | [0.177, 0.270] | [0.298, 0.406] | **[0.346, 0.456]** | [0.031, 0.081] | [0.038, 0.093] | [0.071, 0.139] | [0.063, 0.128] |
| Knorm | [0.003, 0.029] | [0.016, 0.056] | [0.046, 0.105] | [0.046, 0.105] | [0.003, 0.029] | [0.003, 0.029] | [0.009, 0.043] | [0.016, 0.056] |
| RKV | [0.016, 0.056] | [0.054, 0.116] | [0.054, 0.116] | [0.046, 0.105] | [0.009, 0.043] | [0.038, 0.093] | [0.031, 0.081] | [0.016, 0.056] |
| SnapKV | **[0.366, 0.477]** | **[0.366, 0.477]** | **[0.366, 0.477]** | [0.317, 0.426] | **[0.079, 0.150]** | **[0.079, 0.150]** | **[0.088, 0.162]** | **[0.071, 0.139]** |
| StreamingLLM | [0.016, 0.056] | [0.038, 0.093] | [0.063, 0.128] | [0.105, 0.184] | [0.016, 0.056] | [0.009, 0.043] | [0.038, 0.093] | [0.054, 0.116] |
| **Method** | **StrategyQA** | | | | **FOLIO** | | | |
| Full | [0.850, 0.921] | | | | [0.308, 0.416] | | | |
| ShadowKV | [0.594, 0.702] | | | | [0.279, 0.385] | | | |
| H2O | [0.709, 0.805] | [0.794, 0.877] | **[0.805, 0.886]** | [0.783, 0.868] | [0.177, 0.270] | [0.308, 0.416] | [0.298, 0.406] | [0.317, 0.426] |
| Knorm | [0.327, 0.436] | [0.493, 0.605] | [0.625, 0.730] | [0.709, 0.805] | [0.016, 0.056] | [0.023, 0.069] | [0.046, 0.105] | [0.097, 0.173] |
| RKV | [0.366, 0.477] | [0.395, 0.507] | [0.584, 0.692] | [0.656, 0.758] | [0.038, 0.093] | [0.054, 0.116] | [0.079, 0.150] | [0.105, 0.184] |
| SnapKV | **[0.783, 0.868]** | **[0.805, 0.886]** | [0.794, 0.877] | **[0.794, 0.877]** | **[0.327, 0.436]** | **[0.366, 0.477]** | **[0.356, 0.466]** | **[0.356, 0.466]** |
| StreamingLLM | [0.195, 0.291] | [0.337, 0.446] | [0.464, 0.576] | [0.636, 0.740] | [0.016, 0.056] | [0.016, 0.056] | [0.038, 0.093] | [0.114, 0.195] |
| **DeepSeek-R1-Distill-Llama-8B** | | | | | | | | |
| **Method** | **GSM8K** | | | | **Math500** | | | |
| Full | [0.646, 0.749] | | | | [0.404, 0.517] | | | |
| ShadowKV | [0.454, 0.566] | | | | [0.289, 0.395] | | | |
| H2O | [0.317, 0.426] | [0.473, 0.586] | [0.564, 0.673] | [0.554, 0.663] | [0.159, 0.249] | [0.260, 0.364] | [0.308, 0.416] | [0.308, 0.416] |
| Knorm | [0.000, 0.013] | [0.063, 0.128] | [0.150, 0.238] | [0.232, 0.333] | [0.000, 0.013] | [0.003, 0.029] | [0.009, 0.043] | [0.038, 0.093] |
| RKV | [0.031, 0.081] | [0.023, 0.069] | [0.105, 0.184] | [0.132, 0.217] | [0.016, 0.056] | [0.031, 0.081] | [0.009, 0.043] | [0.009, 0.043] |
| SnapKV | **[0.667, 0.768]** | **[0.667, 0.768]** | **[0.688, 0.786]** | **[0.667, 0.768]** | **[0.366, 0.477]** | **[0.385, 0.497]** | **[0.356, 0.466]** | **[0.356, 0.466]** |
| StreamingLLM | [0.038, 0.093] | [0.204, 0.302] | [0.337, 0.446] | [0.503, 0.615] | [0.016, 0.056] | [0.063, 0.128] | [0.168, 0.260] | [0.242, 0.344] |
| **Method** | **CSQA** | | | | **OBQA** | | | |
| Full | [0.698, 0.796] | | | | [0.794, 0.877] | | | |
| ShadowKV | [0.159, 0.249] | | | | [0.260, 0.364] | | | |
| H2O | [0.424, 0.536] | [0.667, 0.768] | [0.677, 0.777] | **[0.677, 0.777]** | [0.424, 0.536] | [0.730, 0.823] | **[0.783, 0.868]** | **[0.794, 0.877]** |
| Knorm | [0.031, 0.081] | [0.232, 0.333] | [0.483, 0.596] | [0.605, 0.711] | [0.016, 0.056] | [0.223, 0.323] | [0.513, 0.625] | [0.646, 0.749] |
| RKV | [0.046, 0.105] | [0.079, 0.150] | [0.123, 0.206] | [0.298, 0.406] | [0.046, 0.105] | [0.046, 0.105] | [0.150, 0.238] | [0.270, 0.375] |
| SnapKV | **[0.688, 0.786]** | **[0.677, 0.777]** | **[0.688, 0.786]** | [0.677, 0.777] | **[0.773, 0.859]** | **[0.783, 0.868]** | [0.783, 0.868] | [0.762, 0.850] |
| StreamingLLM | [0.023, 0.069] | [0.105, 0.184] | [0.298, 0.406] | [0.444, 0.556] | [0.046, 0.105] | [0.114, 0.195] | [0.270, 0.375] | [0.464, 0.576] |
| **Method** | **ReClor** | | | | **DROP** | | | |
| Full | [0.454, 0.566] | | | | [0.105, 0.184] | | | |
| ShadowKV | [0.223, 0.323] | | | | [0.063, 0.128] | | | |

Table 14: Confidence intervals (continued)...

| Budget | 128 | 256 | 384 | 512 | 128 | 256 | 384 | 512 |
|---|---|---|---|---|---|---|---|---|
| H2O | [0.016, 0.056] | **[0.054, 0.116]** | **[0.186, 0.281]** | **[0.327, 0.436]** | [0.038, 0.093] | [0.046, 0.105] | [0.071, 0.139] | [0.079, 0.150] |
| Knorm | [0.000, 0.013] | [0.000, 0.013] | [0.009, 0.043] | [0.071, 0.139] | [0.000, 0.013] | [0.003, 0.029] | [0.003, 0.029] | [0.031, 0.081] |
| RKV | [0.023, 0.069] | [0.016, 0.056] | [0.016, 0.056] | [0.079, 0.150] | [0.016, 0.056] | [0.016, 0.056] | [0.031, 0.081] | [0.046, 0.105] |
| SnapKV | **[0.444, 0.556]** | **[0.054, 0.116]** | [0.031, 0.081] | [0.031, 0.081] | **[0.132, 0.217]** | **[0.114, 0.195]** | **[0.114, 0.195]** | **[0.123, 0.206]** |
| StreamingLLM | [0.000, 0.013] | [0.000, 0.013] | [0.003, 0.029] | [0.038, 0.093] | [0.009, 0.043] | [0.009, 0.043] | [0.063, 0.128] | [0.097, 0.173] |

| Method | StrategyQA | | | | FOLIO | | | |
|---|---|---|---|---|---|---|---|---|
| Full | | [0.688, 0.786] | | | | [0.414, 0.527] | | |
| ShadowKV | | [0.751, 0.841] | | | | [0.279, 0.385] | | |
| H2O | [0.204, 0.302] | **[0.636, 0.740]** | **[0.719, 0.814]** | **[0.740, 0.832]** | [0.046, 0.105] | [0.317, 0.426] | [0.356, 0.466] | **[0.404, 0.517]** |
| Knorm | [0.038, 0.093] | [0.308, 0.416] | [0.513, 0.625] | [0.646, 0.749] | [0.000, 0.013] | [0.016, 0.056] | [0.079, 0.150] | [0.168, 0.260] |
| RKV | [0.054, 0.116] | [0.298, 0.406] | [0.444, 0.556] | [0.574, 0.683] | [0.016, 0.056] | [0.054, 0.116] | [0.097, 0.173] | [0.214, 0.312] |
| SnapKV | **[0.625, 0.730]** | [0.605, 0.711] | [0.584, 0.692] | [0.625, 0.730] | **[0.404, 0.517]** | **[0.395, 0.507]** | **[0.434, 0.546]** | **[0.404, 0.517]** |
| StreamingLLM | [0.016, 0.056] | [0.079, 0.150] | [0.308, 0.416] | [0.503, 0.615] | [0.000, 0.013] | [0.003, 0.029] | [0.023, 0.069] | [0.063, 0.128] |

