# OpenReview forum: "Hold Onto That Thought: Assessing KV Cache Compression On Reasoning"
_ICLR.cc/2026/Conference — Submitted to ICLR 2026_

### Official Review · Reviewer_bPiV · 2025-10-31

**Soundness:** 2
**Presentation:** 2
**Contribution:** 3
**Rating:** 4
**Confidence:** 4

**Summary:**

This paper evaluates a variety of state-of-the-art KV cache compression algorithms across eight prominent reasoning benchmarks. The study fills a gap in existing literature by shifting focus from prefill-stage, prompt-centric compression to long-decoding, multi-step reasoning scenarios, revealing nuanced tradeoffs between memory savings, computational latency, and answer quality.

**Strengths:**

1. The study rigorously benchmarks multiple leading KV cache compression strategies using a representative suite of reasoning datasets
2. The analysis highlights that, contrary to some existing assumptions, attention-based heavy-hitter strategies (H2O, SnapKV-D) dominate in the decoding phase for reasoning models—sometimes even outperforming the full cache baseline. This finding has practical consequences for both research and deployment.

**Weaknesses:**

1. Several directly relevant methods in the recent literature, especially regarding head selection, layer- or depth-wise compression, and low-rank approaches, are omitted from the empirical comparison and insufficiently discussed.
[1] R-KV: Redundancy-aware KV Cache Compression for Reasoning Models
[2] Layer-Condensed KV Cache for Efficient Inference of Large Language Models
2. The mathematical treatment of token importance measures (accumulated attention, L2norm...) is mostly descriptive and empirical, with little formal analysis of conditions under which these heuristics succeed or fail. Beyond Equation 1 and qualitative descriptions, there is limited exploration of convergence, stability, or guarantees about quality degradation as budget shrinks. For example, there is no explicit loss/objective formalization relating retained tokens’ attention scores to model output fidelity
3. Inconsistent or ambiguous presentation. Several figures lack legends or use color encodings without a keyed explanation (see Figure 4), reducing interpretability. Table 1 is dense and difficult to parse, with cryptic abbreviations (e.g., M, DQ, DL, LN) left undefined in the immediate context, and missing values are not clarified. This impedes reproducibility and transparency.
4. The study benchmarks only on a handful of popular reasoning datasets, many of which have similar format and chain-of-thought prompting artifacts. Thus, the generality of claims about method superiority might be weaker in other settings (e.g., dialogue, scientific QA).
5. the core contributions are predominantly evaluative or minor variants (e.g., SnapKV-Decoding). There is no clearly articulated new compression principle or learned policy.

**Questions:**

See Weaknesses for details.

---

> ### Author Response · Authors · 2025-11-21
> **Author Response**
>
> We thank the reviewer for their detailed feedback. We are encouraged that they find our experimental design rigorous and our findings of practical use for the research community. Below, we address all questions and concerns.
>
> >Alternative eviction paradigms.
>
> Thanks for these suggestions. We have added R-KV numbers to Tables 1-4 across all datasets. This reasoning-trace targeting method also seems to lag H2O and SnapKV. For a layer-by-layer strategy, we tested the popular PyramidKV method on GSM8K and MATH500 with results reported in Appendix A.7. We only tested on these two challenging benchmarks since PyramidKV can regarded as a layer-by-layer variation of SnapKV. We find that SnapKV-D is superior to PyramidKV.
>
> >Mathematical treatment of token importance measures
>
> We now add a dedication section on compression objective in Section 2: Preliminaries. Guarantees are rare -- to the best of our knowledge, the most well-known guarantee in the eviction space is due to the heavy-hitter Scissorhands work (Liu et al., 2023). Their Theorem 4.1 measures the norm deviation between the output of a vanilla transformer layer using a full cache and KV compressed layer which evicts tokens with the lowest accumulated attention scores. The heavier-tailed the attention distribution and larger the budget, the less deviation. This result is readily applicable to a vanilla version H2O (no automatic sink and recent context retention). For SnapKV (another heavy-hitter method), a guarantee likely resembles the Scissorhands/H2O result, but the observation window poses an additional complication for analysis which we do not explore here. Our other tested methods KNorm, PyramidKV, SnapKV, and StreamingLLM are unguaranteed and base their eviction strategies based on empirical observations.
>
> > Inconsistent or ambiguous presentation.
>
> Thanks for this suggestion -- we have broken up Table 1 into 4 tables by model to increase readability (there are no longer model abbreviations), improved captions, and cell highlighting. We are open to further stylistic suggestions.
>
> > Method superiority in non-single query settings
>
> We agree that dialogue/multi-turn long-reasoning is a practically relevant, real-world scenario. However, we argue that cache eviction strategies are only appropriate for single-query scenarios by design. Assuming the model cannot see future questions ahead of time, past evicted tokens which are relevant to follow-up queries can never be recovered, therefore we can only anticipate poor performance. Multi-turn datasets are best assessed on KV offloading strategies (ShadowKV, Infingen, MagicPig, Quest, etc.) which maintain the full cache on the CPU. To the best of our knowledge, Figure 7 in the ShadowKV paper (Sun et al., 2025) is the only work* which compares against eviction methods on a multi-turn benchmark (a custom version of NIAH), where
> and StreamingLLM immediately crash on turn 2 (as expected).
>
> That said, we believe that this should encourage further research into token eviction strategies which maintain compressed information regarding evicted tokens. Until then, dialogue seems to only be informative for offloading algorithms.
>
> *StreamingLLM (Xiao et al., 2024) reports high "multi-round" performance in Section 4.3, but for their dataset, all questions and relevant context are passed together as one large prompt, which does not simulate interactive multi-turn Q&A, hence the discrepancy in performance seen in Sun et al., 2025.
>
> > The core contributions are predominantly evaluative or minor variants.
>
> Indeed, our paper is an assessment benchmark, but the long-reasoning scenario was previously under-studied in the token eviction space. Our contributions should be valuable for the research community: (1) revealing that classical methods such as H2O and SnapKV are highly-effective should encourage further algorithmic updates to these strategies. (2) our discovery that smaller budgets can increase talkativity should encourage practitioners to balance memory costs with inference costs. At a high-level, every eviction method down-scales within-layer attention costs to the size of the budget, but we have revealed that aggressive compression may lead to more total computation. We have additionally summarized practical guidance in Section 4.7.
>
> We thank the reviewer again for their comments. If you believe that we have sufficiently addressed your concerns, we do kindly ask that you may re-consider your scoring in light of our updates. We are always open to further discussion as well.

---

### Official Review · Reviewer_N3yv · 2025-11-01

**Soundness:** 2
**Presentation:** 3
**Contribution:** 2
**Rating:** 4
**Confidence:** 3

**Summary:**

The paper examines how different KV cache compression techniques affect the performance of LLMs on reasoning tasks. It highlights that existing compression strategies are often tailored for the prefill phase and may not be suitable for tasks requiring extensive decoding. Through benchmarking, the authors identify H2O and a variant of SnapKV as effective strategies for reasoning models, emphasizing the need for heavy-hitter tracking in reasoning traces. The study also discusses the balance between cache size and inference costs, noting that smaller cache budgets can result in longer reasoning traces.

**Strengths:**

1. Addresses a critical issue in LLMs related to memory constraints during reasoning tasks.
2. Provides a comprehensive evaluation of multiple KV cache compression strategies across various reasoning benchmarks.
3. Identifies specific strategies (H2O and SnapKV variant) that enhance performance in reasoning tasks.

**Weaknesses:**

1. The study focuses on a single LLM (Llama-3.1-8B-Instruct), which may limit the generalizability of the findings to other models.
2. Lacks a discussion on the computational overhead introduced by implementing the recommended compression strategies.
3. Does not provide detailed explanations of the underlying mechanisms of the identified effective strategies.

**Questions:**

1. How do the identified compression strategies perform on LLMs with different architectures or sizes?
2. What are the computational costs associated with implementing H2O and the SnapKV variant in practice?
3. Could the findings be influenced by specific hardware configurations, and how might they generalize across different platforms?
4. Are there any potential drawbacks or limitations to using these compression strategies in real-world applications?

---

> ### Author Response · Authors · 2025-11-21
> **Author Response**
>
> Thank you for your feedback. We are encouraged that you believe our work has identified a critical issue for constrained LLM-inference and that our evaluation is comprehensive. We address all your questions and concerns below.
>
> >The study focuses on a single LLM (Llama-3.1-8B-Instruct), which may limit the generalizability of the findings to other models.
>
> We only study a single non-reasoning model, but the focus of this paper is on reasoning models which produce long traces so **we extensively studied 4 popular reasoning models (Tables 1-4, Figure 4, etc.)**: DeepSeek-R1-Distill-Llama-8B, Llama-3.1-Nemotron-Nano-8B-v1, DeepSeek-R1-Distill-Qwen-7B, and DeepSeek-R1-Distill-Qwen-14B. Our overall trends (superiority of heavy-hitter methods, budget/token output tradeoff) generalize across these architectures.
>
> >Computational overhead
>
> We performed a latency experiment in Figure 3 and have now added end-to-end throughput (tokens/s) numbers in Table 5. We have also added a new section Appendix A.9 documenting the computational complexity of each method for assessing token importance.
>
> >Underlying mechanisms of the identified effective strategies.
>
> We have added further fine-grained analysis in Appendix A.4 studying critical token retention. Put simply: the best methods (H2O and SnapKV-D) focus on high accumulated attention scores as a token importance heuristic. In Figure 4, there is a clear correlation between attention loss and performance (performance increased with better preservation of the attention distribution). Since keywords occupy a high ratio of the final KV cache for SnapKV-D and H2O this indicates that these tokens maintain high importance through decoding, which is empirically well-supported by the Persistence of Importance Hypothesis (Scissorhands, Liu et al., 2023) for long prompt pre-filling, and now confirmed for long decoded reasoning.
>
> >How do the identified compression strategies perform on LLMs with different architectures or sizes?
>
> As demonstrated in our experimental analysis (Tables 1-4), while mean performance varies by reasoning architecture (DeepSeek-R1-Distill-Llama-8B, Llama-3.1-Nemotron-Nano-8B-v1, DeepSeek-R1-Distill-Qwen-7B), method rankings do not vary. This holds true for larger models as well (Table 6) and max token settings (Figure 7).
>
> >What are the computational costs associated with implementing H2O and the SnapKV variant in practice?
>
> We describe these costs with Appendix A.9. Heavy-hitter methods cost more since they need to compute attention to estimate token importance.
>
> >Could the findings be influenced by specific hardware configurations, and how might they generalize across different platforms?
>
> We would expect raw latency and throughput numbers would change between hardware configurations, but we would not expect method rankings (in terms of efficiency) to change. Hardware and kernels designed specifically for fast attention computations would improve the latency/throughput of H2O and SnapKV. The reported accuracies may shift slightly by platform depending on GPU kernels (due to floating point nondeterminism, see He et al., 2025), though this only subtly changes the core content of the output.
>
> >Are there any potential drawbacks or limitations to using these compression strategies in real-world applications?
>
> Token eviction should only be used when inference is performed entirely on a memory-constrained device such as a GPU. Eviction can induce increased talkativity (as evidenced by Figure 4) and introduces additional computational overhead due to token importance estimation (it is well known that vanilla, uncompressed autoregression can be faster). Furthermore, we find that uncompressed models consistently outperform eviction strategies in Tables 1-4.
>
> We thank the reviewer again for their comments. If you believe that we have sufficiently addressed your concerns, we do kindly ask that you may re-consider your scoring in light of our updates. We are always open to further discussion as well.

---

### Official Review · Reviewer_P3nz · 2025-11-01

**Soundness:** 2
**Presentation:** 2
**Contribution:** 2
**Rating:** 4
**Confidence:** 4

**Summary:**

This paper studies KV-cache compression strategies for long-chain reasoning in LLMs. The authors benchmark multiple KV eviction techniques—including H2O, StreamingLLM, K-Norm, and an adapted variant of SnapKV (extended to decoding phase, termed SnapKV-D) on multi-step reasoning tasks. They show that long-form reasoning workloads differ substantially from long-context inference: compression during decoding is critical, and conventional prefill-focused KV strategies can degrade or even destabilize reasoning traces. SnapKV-D generally performs best among tested methods, and the authors also surface an interesting phenomenon where aggressive eviction may increase generation length due to reasoning loops.

**Strengths:**

1. This paper focuses on an increasingly important setting: long reasoning LLMs, where decode-phase KV dominates.
2. Systematic comparison across multiple KV compression methods and budgets.
3. Empirical evidence that heavy-hitter approaches (SnapKV-D, H2O) outperform naive recency or norm methods for reasoning.

**Weaknesses:**

1. The paper evaluates eviction‐based strategies but does not compare against modern sparse attention approaches designed for long reasoning (e.g., DeepSeek sparse attention, SeerAttention). These are important reference points for understanding where eviction sits in the design space.
2. No evaluation on multi-turn or interactive reasoning. Long decoding commonly occurs in iterative workflows (MathChat-style step reasoning, planning with feedback). Single-turn benchmarks may not fully reflect practical long-chain settings.
3. The work observes several phenomena (e.g., compression can increase output length), yet provides limited guidance for practitioners on how to choose or tune strategies in practice.
4. Lack of fine-grained analysis on RL-trained reasoning models. RL-distilled reasoners (e.g., DeepSeek-R1-Distill) have been shown to exhibit different attention concentration patterns vs non-reasoning models. The paper observes performance differences but does not probe whether attention sparsity, focusing behavior, or step-critical token retention varies across training paradigms. Understanding this distinction would increase interpretability and generalization of findings.

**Questions:**

1. Could you elaborate on SnapKV-D?

---

> ### Author Response · Authors · 2025-11-21
> **Author Response**
>
> We thank the reviewer for their valuable feedback. We are motivated that you find our problem setting important, our methodology systematic, and our empirical observations useful. Below, we address lingering questions and concerns.
>
> > Modern sparse attention approaches
>
> We now conduct an analysis against SeerAttention-DeepSeek-R1-Distill-Qwen-14B in Appendix A.8. We do not at this present time compare against DeepSeek NSA as it is currently only available for DeepSeek-V3.2-Exp, a 635B param model.
>
> We do consider the mentioned attention-sparse methods as orthogonal to KV eviction. There are two key differences which make them challenging to compare: (1) Memory: The core focus of every eviction method is **maintaining a reduce, fixed cache size** by permanently throwing away tokens. DeepSeek's NSA and SeerAttention **maintain the full cache** and train learned projections to identify a sparse set of each tokens at every decoding step. (2) Computational overhead: although the secondary benefit of token eviction is sparse attention, in fact, these methods tend to not scale well against naive, uncompressed autoregression because there is overhead involved with calculating token importance for every eviction method. This is known to practictioners -- eviction is only encouraged if memory usage is hard-constrained.
>
> That said, we do agree that in the efficiency design space, it is interesting to compare overall performance against methods using uncompressed caches -- which is why we initially performed a baseline comparison against ShadowKV.
>
> >No evaluation on multi-turn or interactive reasoning.
>
> We absolutely agree that long-chain multi-turn reasoning is a practically relevant, real-world scenario. However, we argue that cache eviction strategies are only appropriate for single-query scenarios by design. Assuming the model cannot see future questions ahead of time, past evicted tokens which are relevant to follow-up queries can never be recovered, therefore we can only anticipate poor performance. Multi-turn datasets are best assessed on KV offloading strategies (ShadowKV, Infingen, MagicPig, Quest, etc.) which maintain the full cache on the CPU. To the best of our knowledge, Figure 7 in the ShadowKV paper (Sun et al., 2025) is the only work* which compares against eviction methods on a multi-turn benchmark (a custom version of NIAH), where SnapKV and StreamingLLM immediately crash on turn 2 (as expected). That said, we believe that this should encourage further research into token eviction strategies which maintain compressed information regarding evicted tokens.
>
> *StreamingLLM (Xiao et al., 2024) reports high "multi-round" performance in Section 4.3, but for their dataset, all questions and relevant context are passed together as one large prompt, which does not simulate interactive multi-turn Q&A, hence the discrepancy in performance as reported in Sun et al., 2025.
>
> >Limited guidance for practitioners
>
> We have now included practical guidance in Section 4.7 bolstered further fine-grained analysis (critical token retention, SnapKV-D window size ablation, etc.).
>
> >Fine-grained analysis
>
> Thank you for this suggestion. We have now included an important token retention study in Appendix A.4. Figure 5 indicates that preserving heavy-hitters correlates with performance, but we further sought to verify that SnapKV-D and H2O were truly maintaining critical tokens more often. We performed an experiment on GSM8K. Questions in this dataset very clearly contain keywords essential for reasoning (name, quantity, object, etc.). We assess how much of the final cache (i.e., after reasoning is complete) contains these keywords. We observe that SnapKV-D and H2O fill the cache with a higher proportion of important keywords for reasoning models, **indicating that keywords maintain high accumulated-attention throughout reasoning.** While this was known for long-prompt pre-filling (see the Scissorhands "Persistence of Importance Hypothesis,"" Liu et al., 2023), it ostensibly holds true for long-decoding as well. For our non-reasoning model this is less apparent between methods.
>
> >Could you elaborate on SnapKV-D?
>
> Yes, we have added further details to Section 4.1 setup and Appendix A.11. To summarize: the principle is very similar as the vanilla SnapKV, but we now allow the observation window to slide along the decoded sequence. First, we select a window size (we choose the recommended size $w=128$). Pre-fill proceeds as normal: a window is placed at the end of the prompt and evicting tokens which produce high-attention against the widow tokens until the budget is met. (If the prompt length is shorter than the budget, we keep everything.) In decoding, we perform SnapKV evictions every $w=128$ steps. The window is placed on the 128 most recent tokens and we evict until the budget is met. In Appendix A.8, we find that window size has negligible effects on performance other than at small budgets.

---

### Official Review · Reviewer_NyP6 · 2025-11-03

**Soundness:** 3
**Presentation:** 3
**Contribution:** 3
**Rating:** 6
**Confidence:** 3

**Summary:**

This paper presents a systematic study of KV cache compression strategies on reasoning-heavy LLM benchmarks, emphasizing the decoding phase rather than the prefill phase. It provides valuable insights into how different eviction methods—particularly attention-based ones like H2O and a decoding-enabled variant of SnapKV (SnapKV-D)—perform under constrained memory settings. The authors also extend NVIDIA’s kvpress library to support decoding-phase compression.

The paper fills a meaningful empirical gap and is well-executed in its benchmarking methodology. While it is not theoretically novel, it is well-motivated and will likely interest both practitioners and researchers working on efficient inference for reasoning-capable LLMs.

**Strengths:**

1. Clear empirical gap: Most cache compression work targets long-prompt tasks (e.g., LongBench, RULER), but this paper focuses on long-reasoning scenarios where the generation dominates memory use. This is a genuinely underexplored regime.

2. Comprehensive evaluation: The benchmark suite spans eight reasoning datasets (DROP, ReClor, FOLIO, StrategyQA, CommonSenseQA, OpenBookQA, GSM8K, MATH-500) and multiple models (Llama-3.1-8B-Instruct, DeepSeek-R1-Distill-Llama/Qwen, Nemotron-Nano-8B). This breadth makes the results broadly relevant.

3. Methodological rigor: The experiments cover multiple cache budgets (128–512), token limits (up to 2048), and both latency and accuracy metrics. The discovery that smaller cache budgets can paradoxically lead to longer reasoning traces is particularly insightful.

4. Practical contribution: Extending kvpress to handle decoding-phase compression is a useful engineering effort likely to be adopted by others studying efficient inference.

5. Empirical insight: Attention-based “heavy-hitter” methods (H2O, SnapKV-D) consistently outperform other approaches. In several settings, SnapKV-D even surpasses full-cache accuracy, showing that selective eviction can regularize reasoning behavior.

**Weaknesses:**

1. Limited novelty: The work is primarily empirical. Extending SnapKV for decoding and modifying kvpress are incremental, though valuable, contributions.

2. Implementation details under-specified: The paper lacks detail on how token importance is updated during decoding for SnapKV-D—e.g., whether attention aggregation happens online or at fixed intervals.

3. Normalization and fairness: Because compression methods produce different output lengths, accuracy comparisons may not be normalized by generation length. This could bias metrics toward “talkative” models.

4. Presentation overload: Table 1 is extremely large and difficult to interpret. Averaging across datasets or grouping results visually (as in Figure 4) would make trends clearer.

5. Experimental gaps: No statistical significance or confidence intervals are reported. Additional ablations (e.g., observation window size, eviction frequency) would better characterize SnapKV-D behavior.

**Questions:**

See the weakness.

---

> ### Author Response · Authors · 2025-11-21
> **Author Response**
>
> Thank you, Reviewer NyP6, for your valuable feedback. We are pleased that you find the empirical gap addressed by our work valuable, our contributions practically useful for the compression community, and our methodology rigorous.
>
> We address your weakness and questions below.
>
> >"Novelty[...] incremental, though valuable, contributions."
>
> Indeed, the focus of this paper is empirical in nature, though as you have hinted we strongly believe the research community will find our discoveries useful. There is a swelling urge to design interesting, new techniques such as R-KV (Cai et al., 2025) which target the unique, repetitive structure of thinking traces, but uncovering that pre-thinking-model heavy-hitter methods are equipped to handle reasoning compression should renew interest in these classical approaches.
>
> >SnapKV-D implementation details...
>
> We have now clarified implementation details for SnapKV-D in Section 4.1 setup and Appendix A.6. Prefill proceeds as normal, using the author-recommend observation window size of 128. During the decode phase, we apply SnapKV-D at regular intervals (corresponding to the window size). Initially, we may not evict any tokens since we are still filling up the budget, but once the budget is reached, every 128 steps, we apply SnapKV-D and evict until the budget is satisfied.
>
> We also performed an ablation on window size effect (which is minimal) in Appendix A.8. Thus, users can opt for the default.
>
> >Output length normalization and fairness
>
> In fact, we consider 'talkativity' a measurable dimension and feature of each compression method which has not be considered in prior literature. As you have noticed, each method's selection of tokens to evict will ultimately affect the total reflection/chain-of-thought length. Our normalization therefore is by max output length ($n=2048$). Our reasoning: the mean output length of all tested models (without compression) is well below this max cutoff, so we are challenging each compression strategy to coerce the model into an answer before the max token limit is reached. Compression strategies which induce high talkativity risk the model not outputting an answer within the token limit, which we accordingly penalize. (We do ablate over max token budget as well in Figure 4.)
>
> >Confidence intervals
>
> Confidence intervals for our core benchmarking results are now reported in Appendix A.13 over 3 runs.
>
> >Window size/eviction frequency
>
> Now reported in Section 4.1 setup and Appendix A.6. Our window size determines the eviction frequency (short window = more applications of SnapKV-D). We do not observe any significant differences, though at small budgets, smaller window sizes experience improved performance. This is intuitive -- while smaller budgets force the model to discard more tokens overall, a smaller window size forces re-assessment of the current cache against recent context more frequently hence improving the overall quality of tokens kept.
>
>
>
> We thank the reviewer again for their comments. If you believe that we have sufficiently addressed your concerns, we do kindly ask that you may re-consider your scoring in light of our updates. We are always open to further discussion as well.

---

### Author Response · Authors · 2025-12-03
**Summary of Rebuttal**

Dear Area Chair:

We thank the reviewers (NyP6, P3nz, N3yv, bPiV) for their feedback. Their remarks have prompted us to conduct additional experiments and analyses, which we believe strengthen the contributions of our work. Although the reviewers did not get a chance to respond to our rebuttal, we summarize the discussion phase below.

**Highlighted strengths by the reviewers**
 - All reviewers noted we captured an important empirical gap. NyP6: "meaningful empirical gap [...]," P3nz: "increasingly important setting [...]," N3yv: "addresses a critical issue in LLMs[...]," bPiV: "study fills a gap [...]"
 - All reviewers noted that we conducted comprehensive experiments.
 - Practical contributions by revealing, unexpectedly, that lower budgets can increase total output length (NyP6, N3yv, bPiV) and that accumulated attention is a dominant importance heuristic for reasoning tokens (all reviewers). Our experiments should serve as a field guide for eviction-based compression of reasoning traces for researchers.

**Concerns and questions addressed by our revision & rebuttal**
 - SnapKV-D implementation (NyP6, P3Nz): now clarified in Section 4.1. Furthermore, we conduct an ablation as requested on SnapKV window sizes in Appendix A.6, revealing that it is not sensitive to these hyperparameters.

 - Readability of Table 1: now broken up into 4 distinct tables by models with improved captions.

 - Comparison to orthogonal trained sparse attention (P3nz): we now compare against SeerAttention as suggested in Appendix A.8.

 - Guidance for practitioners based on our observations (P3nz): now included in Section 4.7, including references to specific Tables/Figures to support each piece of guidance.

 - Fine-grained analysis (P3nz): we now include analysis of step-critical token retention between methods in Appendix A.4, revealing that heavy-hitter methods do, indeed, fill the cache with a higher density of important keywords.

 - Multi-turn scenarios (P3nz, bPiV): we argue that multi-turn/interactive dialogue is rarely studied in token eviction literature, as offloading methods are intuitively superior to eviction strategies, which have been discussed in the InfiniGen paper (Lee et al., 2024).

 - Tested model breadth (N3yv): We clarify that we tested four distinct reasoning models in nearly all of our experiments and conducted an experiment over larger models as well.

- Lack of study on the recent R-KV and layer-wise compression (P3nz): we have now added R-KV to our major study (Tables 1-4). We further note that we have added the implementation of this method to our fork of kvpress. For layer-wise compression, we have added a study of the popular PyramidKV algorithm to Appendix A.7.

 - Confidence intervals (NyP6): now include our largest experiment (Tables 1-4) in A.13.

 - Description of computational overhead for these strategies (N3yv): Asymptotic costs now summarized in Appendix A.9 and end-to-end throughput (tokens/s) have now been added to Table 5.

 - Mathematical discussion of algorithm stability/convergence (bPiV): we have now added discussion to Section 2 discussing the quality of heavy-hitter-compressed outputs through the lens of the celebrated Scissorhands (Liu et al., 2023).

 - Depth of novelty: as stated above, all reviewers have agreed that this paper comprehensively studies a glaring empirical gap in the literature, which is the core value of this extended study. This work should function as a guide for KV compression of long reasoning phases. Furthermore, although extending SnapKV to the decoding phase is not a "difficult" feat of engineering, this modification revealed that a classical method can prove to be very dominant for reasoning compression, which was previously not known. Reviewer NyP6 specifically refers to this as an "incremental, though valuable, contribution." We further note that our included library enables a decoding-based compression study of nearly 20+ eviction strategies.

We strongly believe that we have addressed all key reviewer questions and concerns either through further experiments and/or improved discussion within our revision.

Best regards,

The Authors of Submission #15739

---

### Meta-Review · Area_Chair_voe1 · 2026-01-06

**Summary:**

All reviewers agreed that the paper explores a relevant and timely setting, i.e., KV cache compression during long decoding-phase reasoning (which has been less studied compared to long-prompt prefilling). They acknowledged the breadth of the experimental evaluation and found several empirical observations interesting, particularly the effectiveness of heavy-hitter methods (e.g., H2O, SnapKV-D) and the counterintuitive increase in reasoning length under aggressive compression.

However, several concerns were raised by reviewers as well. Multiple reviewers emphasized that the paper’s contributions are largely empirical and/or incremental, lacking a clear novel compression principle or algorithmic insight beyond benchmarking and minor variants of existing methods (Reviewers NyP6 and bPiV). Several key aspects were initially under-specified or insufficiently analyzed, (e.g. implementation details of SnapKV-D), although these are more easily addressable in a minor revision. Reviewers also noted missing or limited comparisons in the original paper to closely related and more recent approaches, such as sparse attention mechanisms, redundancy-aware eviction (e.g., R-KV), layer-wise compression, etc, which weakens the paper’s positioning in the broader efficiency landscape (Reviewers P3nz and bPiV). While the authors provided some comparisons during the rebuttal period, for a thorough evaluation, these results (and any other relevant associated methods) should be incorporated in the main paper and ideally evaluated in another round of reviewing.

In addition, I agree with the reviewer concerns remain about the generality of the findings (Reviewers P3nz, bPiV). The study focuses primarily on single-turn reasoning benchmarks, with limited evidence that the conclusions extend to interactive or multi-turn reasoning settings. More specifically, the authors argue that cache eviction strategies are only appropriate for single-query settings, which might as well be correct, but the framing of the paper should reflect that (e.g. in abstract and introduction) as the current statements are quite general assertions.

**Reviewer Concerns:**

Several reviewer concerns were discussed and addressed in the rebuttal. Reviewers NyP6 and P3nz requested clearer SnapKV-D implementation details and eviction frequency, which were addressed through added explanations and ablations. Concerns about statistical rigor raised by Reviewer NyP6 were partially addressed via the inclusion of confidence intervals. Some comparisons to SeerAttention (Reviewer P3nz) and R-KV and PyramidKV (Reviewer bPiV) were also added.

However, several substantive concerns remain outstanding in my view. I agree with reviewers NyP6 and bPiV's primary concern that the core contribution remains largely empirical and incremental, with no fundamentally new compression principle or formal objective introduced. Moreover, the theoretical treatment of token importance and stability remains limited despite added discussion (Reviewer bPiV). In addition, the generality or applicability of the findings (e.g. to multi-turn settings) requires a tempering/reframing of the claims and assertions made in the paper (e.g. there is no mention of only single-turn scenario in the abstract and/or introduction). Overall, while the rebuttal strengthened the work, it did not fully resolve shared concerns outlined above.

**Reviewer Scores:**

- Reviewer NyP6: I believe they would have maintained their score (or even decreased it owing to the outstanding limited novelty concerns).
- Reviewers P3nz and bPiV: I believe they would have maintained their scores as the concerns about generality, i.e., single-turn vs multi-turn scenarios remains outstanding.
- Reviewer N3yv: I believe they would have increased their score to a positive assessment as their concerns were resolved.

---

### Decision · Program_Chairs · 2026-01-26

Reject